# Adversarial Paraphrasing: A Universal Attack for Humanizing AI-Generated Text

**Yize Cheng**[*]  **Vinu Sankar Sadasivan**[*]  **Mehrdad Saberi**[†]  **Shoumik Saha**[†]  **Soheil Feizi**
University of Maryland, College Park
{yzcheng, vinu, msaberi, smksaha, sfeizi}@cs.umd.edu

 **Project**: https://github.com/chengez/Adversarial-Paraphrasing

## Abstract

The increasing capabilities of Large Language Models (LLMs) have raised concerns about their misuse in AI-generated plagiarism and social engineering. While various AI-generated text detectors have been proposed to mitigate these risks, many remain vulnerable to simple evasion techniques such as paraphrasing. However, recent detectors have shown greater robustness against such basic attacks. In this work, we introduce **Adversarial Paraphrasing**, a training-free attack framework that universally humanizes any AI-generated text to evade detection more effectively. Our approach leverages an off-the-shelf instruction-following LLM to paraphrase AI-generated content under the guidance of an AI text detector, producing adversarial examples that are specifically optimized to bypass detection. Extensive experiments show that our attack is both broadly effective and highly transferable across several detection systems. For instance, compared to simple paraphrasing attack—which, ironically, increases the true positive at 1% false positive (T@1%F) by 8.57% on RADAR and 15.03% on Fast-DetectGPT—adversarial paraphrasing, guided by OpenAI-RoBERTa-Large, reduces T@1%F by 64.49% on RADAR and a striking 98.96% on Fast-DetectGPT. Across a diverse set of detectors—including neural network-based, watermark-based, and zero-shot approaches—our attack achieves an average T@1%F reduction of 87.88% under the guidance of OpenAI-RoBERTa-Large. We also analyze the tradeoff between text quality and attack success to find that our method can significantly reduce detection rates, with mostly a slight degradation in text quality. Our adversarial setup highlights the need for more robust and resilient detection strategies in the light of increasingly sophisticated evasion techniques.

## 1  Introduction

Recent advancements in natural language generation have given rise to transformer-based Large Language Models (LLMs) such as GPT [23], Gemini [7], and LLaMA [21], which have demonstrated remarkable capabilities across a wide range of tasks, such as email composition and code generation. These models are capable of producing fluent, coherent text that can be difficult to distinguish from that written by humans. However, despite their impressive performance, LLMs also raise significant security and ethical concerns, including risks related to plagiarism and social engineering.

To counter these risks, the development of reliable AI-generated text detection tools has become a critical research problem. Several works have proposed training neural network-based classifiers to address this challenge [10, 8, 26, 30, 34, 18]. Although typically weaker than trained detectors,

---

[*]Equal contribution
[†]Equal contribution

39th Conference on Neural Information Processing Systems (NeurIPS 2025).

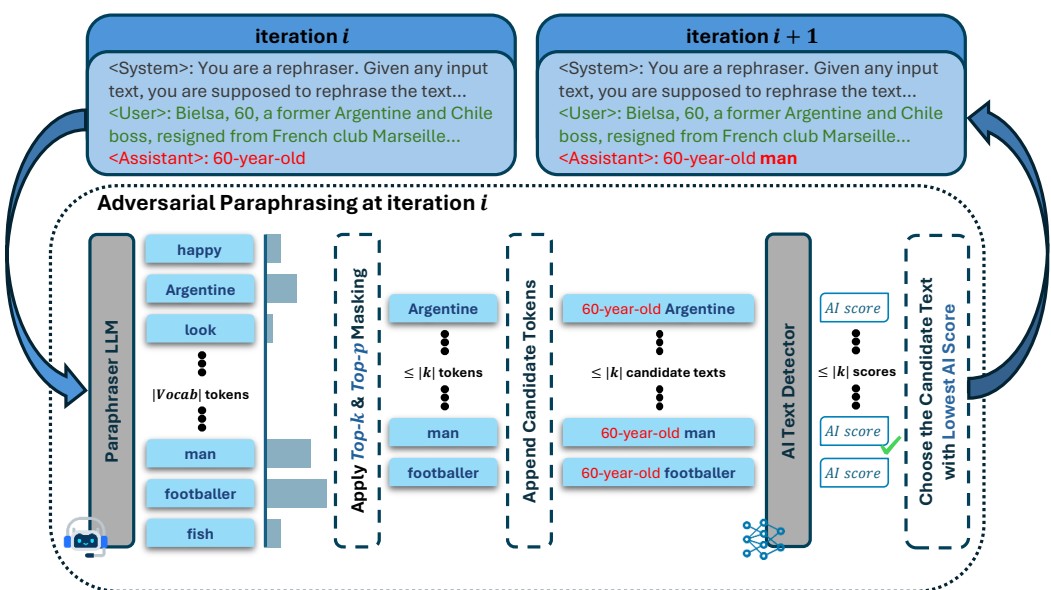

Figure 1: **An overview of our universal and training-free framework for humanizing AI text.** At every auto-regressive step of adversarial paraphrasing, using the guidance from an AI text detector, we search for the token with lowest "AI-score" from the set of top candidate tokens sampled by the paraphraser LLM. The token generation iterations continue until the paraphrasing is finished. (i.e. [EOS] token is sampled)

various zero-shot detection techniques [22, 1, 6, 17] have also been introduced to reduce the overhead of training classifiers. Another propitious approach is watermarking LLMs, which enforces specific signatures in the output text to facilitate detection [2, 14, 16, 36, 4]. While these detection methods show promise in specific contexts, concerns about their robustness remain.

Sadasivan *et al.* [28] and Krishna *et al.* [15] demonstrated how AI-generated text can be paraphrased by another AI model to evade detection by the existing AI detectors. The former also showed that recursively paraphrasing AI-generated content can be used to stress-test these systems and highlighted the fundamental difficulty of reliably detecting AI-generated text. These paraphrasing-based attacks aim to obscure the statistical patterns and artifacts that detection models typically rely on. However, more recent work suggests that some advanced detectors — especially those trained on paraphrased AI outputs [8] — may exhibit greater resilience to such evasion techniques. These developments naturally lead to a pressing question: *"Is it possible to develop a universal attack framework that can consistently and effectively bypass these robust AI-generated text detectors with transferability to a wide variety of other detection systems?"*

In this paper, we introduce a universal, training-free framework for humanizing AI-generated text, a novel attack designed to effectively and efficiently rephrase AI-generated text to evade a wide range of detectors with greater success when compared to the existing attack evasion methods. Our approach, **Adversarial Paraphrasing**, leverages an existing instruction fine-tuned LLM (LLaMA-3-8B configured as a paraphraser via a custom system prompt) not just for paraphrasing, but for adversarially humanizing the text under the guidance of a trained AI text detector (see Figure 1). During the paraphrasing process, at each token generation step, we consider the top likely next tokens proposed by the paraphrasing LLM. Then, instead of standard decoding, we score each of these potential continuations using a guidance AI text detector. We then select the token that leads the sequence to be predicted with the lowest "AI score" (or equivalently, the highest "human score") by the detector, akin to a targeted, detector-guided beam search at depth one. Some example results are shown in Table 1. This method operationalizes the goal of humanization directly within the generation process, falling under the umbrella of controlled text generation [37, 19, 3].

We demonstrate that our adversarial paraphrasing strategy is not only more effective than baseline methods such as simple or recursive paraphrasing, but unlike other methods, it is also universally transferable to other diverse detection methods, including neural network-based, zero-shot, and watermarking-based detectors. This justifies the universality of our framework. For example, while

simple paraphrasing ironically increases true positive rate at 1% false positive rate (T@1%F) by 8.57% on RADAR [8] and 15.03% on Fast-DetectGPT [1], our adversarial approach, guided by OpenAI-RoBERTa-Large [30], reduces T@1%F by 64.49% and a striking 98.96%, respectively. Evaluated across eight detectors from three different categories, our method achieves an average T@1%F reduction of 87.88%, all while maintaining minimal degradation in text quality. We comprehensively analyze the tradeoff between our attack and the text quality using multiple automated studies.

Our core contributions are:

- We introduce **Adversarial Paraphrasing** — a universal and transferable attack that humanizes AI-generated text by guiding an off-the-shelf LLM, used as a paraphraser, to sample each token based on detection scores from an off-the-shelf trained AI text detector.

- We conduct extensive experiments demonstrating the attack's effectiveness and transferability across 8 different types of state-of-the-art AI text detectors, highlighting its universality.

- We evaluate the trade-off between attack success and the resulting text quality using perplexity scores as well as automated ratings with GPT-4o to find that our attacks can significantly reduce detection rates when compared to prior evasion attacks with only a slight or no degradation in the text quality.

Our findings highlight important vulnerabilities of existing AI text detectors in the presence of adversaries.

## 2   Related Work

**AI- Text Detection.** Recent studies have demonstrated the effectiveness of training neural networks to classify AI-generated versus human-written text [10, 8, 26, 30, 34, 18]. For example, Solaiman *et al.* [30] employed a RoBERTa-based [20] classifier to distinguish between human-written and GPT-2-generated texts. Building on this idea, Li *et al.* [18] improves their classifier's performance in the wild by gathering a diverse dataset, MAGE, for training their network. RADAR [8] adversarially trains their detector using a paraphraser in an iterative manner to make their detector robust to paraphrasing attacks. To reduce the computational overhead of training a network-based detector, various zero-shot detectors are proposed [22, 1, 6, 17]. These detectors use an off-the-shelf LLM to evaluate statistical properties of candidate texts, such as their entropy or log probability scores, to perform detection. For example, DetectGPT [22] observes that AI-generated text tends to lie in regions of negative curvature in the log-probability landscape, while Fast-DetectGPT [1] improves efficiency by introducing conditional probability curvature. Watermarking has also been explored as a means of identifying AI-generated text. Kirchenbauer *et al.*[14] introduced the KGW watermark scheme that partitions the token vocabulary into green and red lists, encouraging the model to sample more green tokens, thereby embedding detectable patterns in the generated text. While the KGW scheme uses previously generated tokens to seed a random number generator for token partitioning, Zhao *et. al.* [36] proposed a variant of KGW, the Unigram watermark, which uses a fixed token partitioning for generation for provably demonstrating robustness. Kuditipudi *et al.* [16] developed a robust, distortion-free watermarking technique aimed at reducing distributional shifts caused by watermarking. More recently, Dathathri *et al.* [4] introduced SynthID, which utilizes tournament sampling to create a scalable watermarking solution for LLMs employing speculative decoding.

**Attacks on AI Text Detectors.** Sadasivan *et al.* [28] showed that AI text detectors can be fooled using paraphrasing attacks. While basic paraphrasing methods are sufficient to defeat early zero-shot and trained detectors, more robust detectors [8, 14, 36, 1] require recursive paraphrasing to be effectively bypassed. To this end, Krishna *et al.* [15] proposed DIPPER, a powerful T5-based [24] paraphrasing model that significantly enhances the effectiveness of such attacks. Sadasivan *et al.* [28] also introduced spoofing attacks, where human-written texts are manipulated to be misclassified as AI-generated, thereby increasing type-I errors. In addition, they analyzed the theoretical difficulty of AI text detection, highlighting a fundamental trade-off between type-I and type-II errors even for optimal detectors. The theoretical limits of AI-generated content watermarking have also been explored in other works [35, 27], which analyze the inherent challenges in maintaining robustness under attack. There also exist previous adversarial attack methods for breaking watermarking techniques. For example, Jovanovic *et al.* [12] proposed *Watermark Stealing*, a technique that learns the watermark signatures of a language model and uses this knowledge to both evade and spoof watermark-based detectors effectively. However, this framework specifically targets watermarking detectors and is not

transferable to other text detectors. In this paper, we propose Adversarial Paraphrasing, a training-free attack that can universally break a variety of text detectors without the knowledge of the detection scheme, and that can even break more robust AI detectors [8] trained to withstand simple or recursive paraphrasing attacks.

**Controlled Text Generation.** Dathathri *et al.* [3] proposed the Plug and Play Language Model (PPLM) where an LLM can generate tokens with control at decoding time, guided by various attribute classifiers. They show the effectiveness of their method to generate text guided by various classifiers to switch topics or sentiments. CAT-Gen [33] introduced controllable generation similar to that in PPLM, to adversarially generate diverse, fluent datasets using unrelated attribute classifiers as guidance. PPLM and CAT-Gen use gradient computations to perturb the key-value pairs of the transformer network to steer the generation to favor a selected attribute. This differs from our approach since our adversarial paraphrasing is a gradient-free approach for controlled paraphrasing. InstructCTG [37] demonstrated how off-the-shelf LLMs can controllably generate texts by incorporating the constraints as natural language instructions. InstructCTG, similar to our work, uses verbalized instructions to control the output generations. While they use verbalization to constrain generations lexically or syntactically, we use system prompts to constrain the LLMs we use to behave as a paraphraser. However, InstructCTG requires fine-tuning the LLM on the augmented corpus while we use the power of Instruction-following LLMs to directly use them off-the-shelf without any gradient computations. BEAST [29] proposed inference-time beam search-based guidance to generate adversarial tokens to jailbreak LLMs. BEAST is the most relevant prior work to our paper since they use a gradient-free bi-level beam search approach to find adversarial prompts guided by an adversarial objective function. In contrast, our work uses a gradient-free single-level beam search to find adversarial paraphrases guided by an AI text detector.

## 3   Adversarial Paraphrasing for Universal Humanization of AI Text

In this section, we present our adversarial paraphrasing framework, designed to universally paraphrase AI-generated text to evade detection on various detectors. Algorithm 1 outlines our method, and Figure 1 provides an illustrative visual overview of each step in the iterative paraphrasing process. Our approach auto-regressively generates paraphrased text using a paraphrasing model $\mathcal{P} : \mathcal{X} \to \mathcal{X}$, guided by a neural network-based detector $\mathcal{D} : \mathcal{X} \to [0, 1]$, where $\mathcal{X}$ denotes the space of natural language texts. The model $\mathcal{P}$ outputs the next token logit distribution $p(\cdot|x) \in \mathbb{R}^d$ for any input $x \in \mathcal{X}$, where $d$ represents the vocabulary size. In the standard setting, the paraphrasing model would multinomially sample the next token from this distribution. In our method, however, token selection is influenced by the detector $\mathcal{D}$, which assigns lower AI-scores (i.e., closer to 0) to more "human-like" texts in the eyes of the detector.

---

**Algorithm 1** Adversarial Paraphrasing with Guidance for Universal Humanization of AI Texts

---

**Require:** Paraphraser LLM $\mathcal{P}$ modeled by $p(\cdot|x)$, guidance detector $\mathcal{D}$, tokenizer decode method $\mathcal{T}$
**Input:** System instruction $sys$, AI-generated text $x_{:n}$, top-p and top-k token masking methods $top\_p$ and $top\_k$
**Output:** Humanized text $y_{:m}$
   ▷ Initialize the empty output string
1: $y = $ "", $m = 0$
   ▷ Auto-regressive adversarial paraphrasing loop
2: **while** True **do**
3:    $\mathbf{p} = p(\cdot|sys \oplus x_{:n} \oplus y_{:m})$
4:    $\mathbf{p}' = top\_k \circ top\_p(\mathbf{p})$
      ▷ Decode logits to text tokens
5:    $candidates = \mathcal{T}(\mathbf{p}')$
6:    $scores = [\,]$
      ▷ Score the candidates from the detector for guidance
7:    **for** $k = 1$ **to** length($candidates$) **do**
8:       $scores$.append($\mathcal{D}(y_{:m} \oplus candidates[k])$)
9:    **end for**
10:   $y^* = candidates[\arg \min scores]$
      ▷ Append the candidate text token with lowest "AI score" (or equivalently, the highest "human score")
11:   $y = y \oplus y^*$, $m = m + 1$
12:   **if** $y^* = $ [EOS] **then**
13:      break
14:   **end if**
15: **end while**

---

**Overview of the Algorithm.** As shown in line 1 of Algorithm 1, we initialize the output string as empty, i.e., $y = $ "". The algorithm then enters an auto-regressive loop (lines 2–15), continuing until

the end-of-sentence ([EOS]) token is generated. At each iteration, the paraphraser computes the logit distribution for the next token (line 3). To narrow down the candidate set, we apply top-$p$ filtering to only select the top tokens whose cumulative probability exceeds a certain threshold $p$, and top-$k$ filtering to cap the maximum number of candidate tokens (line 4). In line 5, we decode the filtered logits into their corresponding textual representations using the decoding function $\mathcal{T}$. From lines 6 to 9, we score each candidate by appending it to the current output and evaluating the resulting text using the detector $\mathcal{D}$. The token associated with the lowest detector score is selected and appended to the output sequence (lines 10 and 14). The loop continues until the [EOS] token is generated.

**Paraphrasing Model Setup.** To ensure effective paraphrasing, our framework relies on the availability of a high-quality paraphrasing model. To this end, we design the framework to be compatible with any well-performing instruction-tuned LLM by leveraging customized system instructions. As illustrated in Figure 2, these prompts guide the LLM to behave as a reliable paraphrasing model, ensuring consistent and contextually appropriate paraphrases. This controlled generation approach is inspired by methods such as InstructionCTG [37].

> You are a rephraser. Given any input text, you are supposed to rephrase the text without changing its meaning and content, while maintaining the text quality. Also, it is important for you to output a rephrased text that has a different style from the input text. You can not just make a few changes to the input text. The input text is given below. Print your rephrased output text between tags <TAG> and </TAG>.

Figure 2: The system prompt used to configure our paraphraser LLM.

**Intuition Behind Universal Transferability.** As demonstrated in our experimental results (see Section 4), our attack consistently evades a wide range of unseen AI text detectors. We attribute this transferability to the guidance signal provided by the guidance detector during generation. This signal plays a critical role in shaping the paraphrased text to align more closely with the statistical properties of human-written language learned by the guidance detector.

The key intuition is that most, if not all, high-performing detectors tend to converge toward a common distribution that characterizes human-authored text, in an effort to minimize false positives. Consequently, if a paraphraser is guided to evade detection by a well-trained detector, its outputs may naturally align more closely with this shared human text distribution. As a result, the generated text becomes more difficult to detect not only by the detector used for guidance, but also by other detectors—since they are all ideally calibrated to the same underlying distribution of human-written text. This property makes our adversarial paraphrases broadly transferable across different detectors.

## 4 Experiments

In this section, we present experimental results demonstrating the effectiveness and transferability of adversarial paraphrasing. We first outline our experimental setup in Section 4.1. In Section 4.2, we report our main finding: adversarial paraphrasing guided by a trained detector can successfully evade a variety of detectors, including trained classifiers, watermark-based detectors, and zero-shot detectors, achieving stronger attack effectiveness and universality compared to simple paraphrasing and recursive paraphrasing baselines. We also include a comparison against watermark stealing attacks [12], which is specifically designed to target watermarking techniques, in Appendix A.

### 4.1 Setup

**AI Text Detectors.** To demonstrate the universality and transferability of our attack, we evaluate it against a wide range of AI text detectors. While there are a plethora of open-sourced AI text detectors, we select in total eight popular and representative detectors from each class, including neural network-based detectors (OpenAI-RoBERTa-Base [30], OpenAI-RoBERTa-Large [30], MAGE [18], and RADAR [8]), watermark-based detectors (KGW [14] and Unigram [36] watermarks), and zero-shot detectors (Fast-DetectGPT [1] and GLTR [6]). We refer to the detector used to guide the adversarial candidate text selection in our method as the *guidance detector*, and the detector that is deployed for AI-generated text detection as the *deployed detector*.

**Datasets.** For non-watermark-based detectors, we use MAGE [18] as our primary evaluation dataset due to its rich diversity of text sources. We randomly sample 2000 AI-generated texts and 2000 human-written texts from MAGE while ensuring that each text is ∼100 to 500 tokens in length. For

| Text | Rating |
|---|---|
| **Original AI Text.** There doesn't seem to be a whole lot of information available about DOCSIS 3.1 yet, but if my prior experience can lend a hand towards this question, I would venture The DOCSIS 3.1 issue has a number of things to keep in mind. First, the DOCSIS 3.1 expansion is only available for initial support, and this upgrade is apparently working on [continues. . . ] | – |
| **Simple Paraphrase.** While the available information on DOCSIS 3.1 is somewhat scarce, I'll draw upon my past experience to provide some insights. The DOCSIS 3.1 upgrade, currently only available for initial support, is being tested on [continues. . . ] | 5 |
| **Adversarial Paraphrase.** Given the scarce details currently available about DOCSIS 3.1, my insight garnered from past experience will attempt to shed some light on the matter. From my understanding, DOCSIS 3.1 necessitates several factors to be taken into consideration. Initially, DOCSIS 3.1 upgrades are only accessible through limited channels, specifically [continues. . . ] | 5 |
| **Original AI Text.** No, addiction is much more than simply a habit; it is a chronic and progressive medical disorder. While habits can be formed through repeated use of drugs or alcohol, addiction is a physical and psychological response to those drugs that causes compulsive drug seeking and use despite the negative consequences. It has been suggested that addiction is [continues. . . ] | – |
| **Simple Paraphrase.** Addiction is a far more complex phenomenon than mere habituation; it is a debilitating and relentless medical condition that has a profound impact on an individual's psyche and physiology. While habits can be formed through repeated exposure to substances, addiction is a primal response to those substances, driving an individual to compulsively seek and consume them despite the devastating consequences. Research suggests that addiction is a direct result of [continues. . . ] | 5 |
| **Adversarial Paraphrase.** Addiction goes beyond being simply a habit; it is a persistently progressing medical condition with multifaceted causes. Initially, habits can form as a result of regular drug or alcohol consumption. However, addiction takes it a step further, evolving into an all-consuming compulsion driven by physiological and psychological cravings, despite the harmful effects it wreaks on one's life. Scientific discoveries imply that [continues. . . ] | 5 |
| **Original AI Text.** The occurrence of near misses was linked to socio-economic and clinical factors as well as issues with care, highlighting the critical need for measures to enhance these indicators. Furthermore, we recommend a more effective incorporation of this topic into the medical course curriculum to emphasize the significance of preventing near misses through appropriate medical training [continues. . . ] | – |
| **Simple Paraphrase** To address the pressing issue of near misses, it is essential to consider the complex interplay of socio-economic, clinical, and care-related factors. A crucial step in mitigating this problem is to incorporate a comprehensive approach into medical education, ensuring that healthcare professionals are adequately equipped to prevent near [continues. . . ] | 5 |
| **Adversarial Parphrase.** Ensuring patient safety necessitates addressing socio-economic, clinical, and care-related factors contributing to near misses. To combat these occurrences, it is essential to overhaul the medical curriculum to stress the importance of preventative measures through targeted training [continues. . . ] | 5 |

Table 1: **Examples of original AI texts with their simple and adversarial paraphrases** (guided by OpenAI-RoBERTa-Large [30]). GPT-4o quality ratings are provided for each paraphrased version.

watermark-based detectors, we construct "watermarked" datasets using a watermarked LLaMA-3.1-8B-Instruct [21]. Specifically, we input the model with the first 20 words of each of the 2000 AI texts as prefix, and let it generate watermarked text ∼200 to 600 tokens in length. We report detailed token statistics for all datasets used in Appendix C.

**Attack setup.** We use LLaMA-3-8B-Instruct [21] with a custom system prompt (see Figure 2) as our paraphraser model. During adversarial sampling, we apply top-p and top-k masking with $p = 0.99$ and $k = 50$ at each step. We ablate the guidance detector using all four neural network-based detectors considered in our study—OpenAI-RoBERTa-Large [30], OpenAI-RoBERTa-Base [30], MAGE [18], and RADAR [8].

**Baselines.** As a simple baseline, we use a single round of paraphrasing [28, 15]. We also evaluate against a stronger recursive paraphrasing [28] baseline. Additionally, in Appendix A, we include a comparison with watermark stealing [12], an attack specifically designed to target LLM watermarking.

## 4.2 Effectiveness and Universality of Adversarial Paraphrasing for Humanizing AI Text

Figure 3 presents the ROC curves illustrating the detection performance of eight different deployed detectors with and without various attack methods. We consider four neural network-based detectors, two watermark-based detectors, and two zero-shot detectors for our experiments. Table 2 reports three key evaluation metrics for each combination of attack method and detector: the Area Under the ROC Curve (AUC), the True Positive Rate at 1% False Positive Rate (T@1%F), and GPT-4o's automated assessments of text quality (Rating). Additional details on text quality evaluation are provided in Section 5. Table 1 provides representative examples of original AI, simple paraphrased, and adversarially paraphrased texts to support manual qualitative comparison.

**Effectiveness.** From the ROC curves, we observe that adversarial paraphrasing consistently and significantly reduces detection performance across all evaluated detectors when compared to simple and recursive paraphrasing baselines. Specifically, adversarial paraphrasing shifts the ROC curves closer to, and sometimes even beyond that of a random detector, resulting in a lower AUC and a significant drop in T@1%F. Notably, RADAR [8]—a detector adversarially trained to be robust to paraphrasing attacks—exhibits improved detection rates after baseline simple and recursive paraphrasing attacks. However, adversarial paraphrasing significantly reduces RADAR's detection. This detection degradation post-attack is more pronounced in other detectors, including watermark-

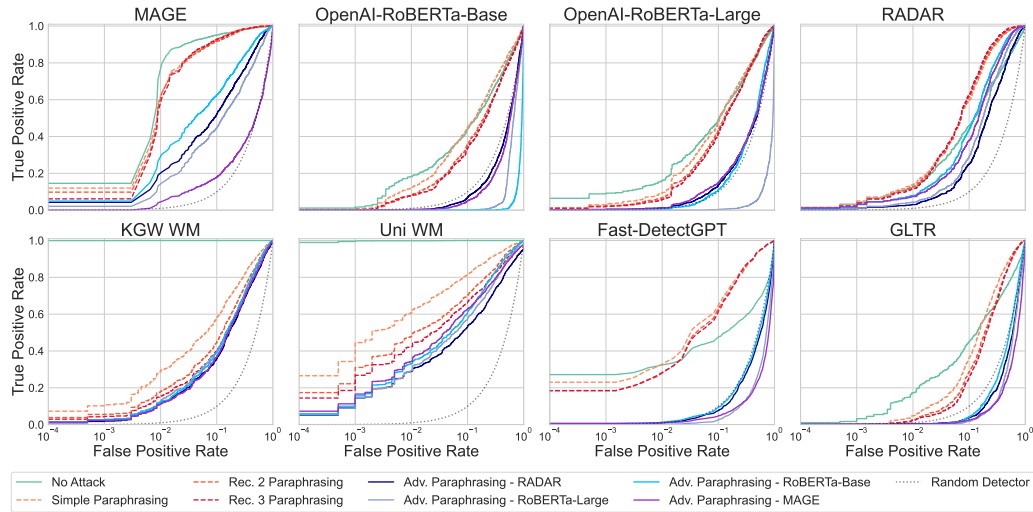

Figure 3: **ROC curves illustrating the AI text detection performance on several deployed detectors, including neural network-based, watermark-based, and zero-shot detectors.** The false positive rate (FPR) axis is displayed in log-scale to highlight fine-grained distinctions in the low-FPR region. It can be observed that adversarial paraphrasing consistently and significantly reduces the detection performance across all deployed detectors when compared to the baselines.

| | RoBERTa-Large | | RoBERTa-Base | | MAGE | | RADAR | | |
|---|---|---|---|---|---|---|---|---|---|
| | AUC ($\downarrow$) | T@1%F ($\downarrow$) | AUC ($\downarrow$) | T@1%F ($\downarrow$) | AUC ($\downarrow$) | T@1%F ($\downarrow$) | AUC ($\downarrow$) | T@1%F ($\downarrow$) | Rating |
| No Attack | 0.789 | 0.163 | 0.745 | 0.182 | 0.975 | 0.768 | 0.767 | 0.124 | – |
| Simple Paraphrase | 0.794 | 0.096 | 0.762 | 0.119 | 0.970 | 0.616 | 0.881 | 0.140 | 4.75 ± 0.54 |
| Rec. Para. 2 | 0.777 | 0.069 | 0.712 | 0.082 | 0.967 | 0.609 | 0.885 | 0.130 | 4.47 ± 0.67 |
| Rec. Para. 3 | 0.779 | 0.059 | 0.706 | 0.079 | 0.969 | 0.585 | 0.893 | 0.117 | 4.26 ± 0.74 |
| AdvPara (RADAR) | 0.538 | 0.013 | 0.464 | 0.004 | 0.815 | 0.201 | 0.723 | 0.031 | 4.45 ± 0.79 |
| AdvPara (RoBERTa-Large) | 0.147 | 0.000 | 0.323 | 0.000 | 0.769 | 0.142 | 0.768 | 0.044 | 4.48 ± 0.77 |
| AdvPara (RoBERTa-Base) | 0.557 | 0.006 | 0.110 | 0.000 | 0.861 | 0.291 | 0.826 | 0.080 | 4.54 ± 0.59 |
| AdvPara (MAGE) | 0.543 | 0.011 | 0.435 | 0.003 | 0.518 | 0.045 | 0.807 | 0.074 | 4.54 ± 0.70 |

| | KGW WM | | Uni WM | | Fast-DetectGPT | | GLTR | | |
|---|---|---|---|---|---|---|---|---|---|
| | AUC ($\downarrow$) | T@1%F ($\downarrow$) | AUC ($\downarrow$) | T@1%F ($\downarrow$) | AUC ($\downarrow$) | T@1%F ($\downarrow$) | AUC ($\downarrow$) | T@1%F ($\downarrow$) | Rating |
| No Attack | 1.000 | 1.000 | 1.000 | 0.999 | 0.666 | 0.323 | 0.726 | 0.174 | – |
| Simple Paraphrase | 0.841 | 0.295 | 0.927 | 0.609 | 0.873 | 0.326 | 0.782 | 0.049 | 4.75 ± 0.54 |
| Rec. Para. 2 | 0.790 | 0.181 | 0.881 | 0.480 | 0.867 | 0.275 | 0.745 | 0.026 | 4.47 ± 0.67 |
| Rec. Para. 3 | 0.762 | 0.155 | 0.858 | 0.424 | 0.867 | 0.276 | 0.739 | 0.025 | 4.26 ± 0.74 |
| AdvPara (RADAR) | 0.741 | 0.117 | 0.777 | 0.291 | 0.452 | 0.009 | 0.433 | 0.004 | 4.45 ± 0.79 |
| AdvPara (RoBERTa-Large) | 0.769 | 0.131 | 0.827 | 0.294 | 0.338 | 0.003 | 0.400 | 0.001 | 4.48 ± 0.77 |
| AdvPara (RoBETa-Base) | 0.769 | 0.125 | 0.852 | 0.332 | 0.480 | 0.012 | 0.481 | 0.001 | 4.54 ± 0.59 |
| AdvPara (MAGE) | 0.750 | 0.113 | 0.831 | 0.344 | 0.301 | 0.011 | 0.338 | 0.003 | 4.54 ± 0.70 |

Table 2: **Detection performance of eight different deployed detectors in distinguishing between AI-generated and human-written text under different attack scenarios.** Metrics reported include AUC and TPR at 1% FPR for each detector. Additionally, we present the mean ± standard deviation of quality ratings given by GPT-4o. Further details on text quality analysis are provided in Section 5.

based and zero-shot detectors. The superior performance of our attack highlights the importance of the guidance signal provided by a trained detector, which effectively steers the paraphrasing to adversarially align with the statistical characteristics of human-authored content.

**Universality.** We find that adversarial paraphrasing guided by one detector can reduce the detection rates of all the other detectors we consider, showing the universal transferability of our method. We also find that any target deployed detector can be evaded by adversarial paraphrasing guided by any trained detectors we consider in our study. We present these findings by plotting the complete transferability matrix in Figure 4, presenting the relative drop in T@1%F for all guidance–deployment combinations of detectors.

Our results show that adversarial paraphrasing is robust to the choice of guidance detectors that we consider in our study. On average, we observe a relative drop in T@1%F of 84.94% when using MAGE as guidance, 86.89% with RADAR, 80.75% with OpenAI-RoBERTa-Base, and 87.88% with OpenAI-RoBERTa-Large. Although different guidance detectors may yield slightly varying degrees of transferability depending on the deployed detector, all our adversarially paraphrased outputs, agnostic

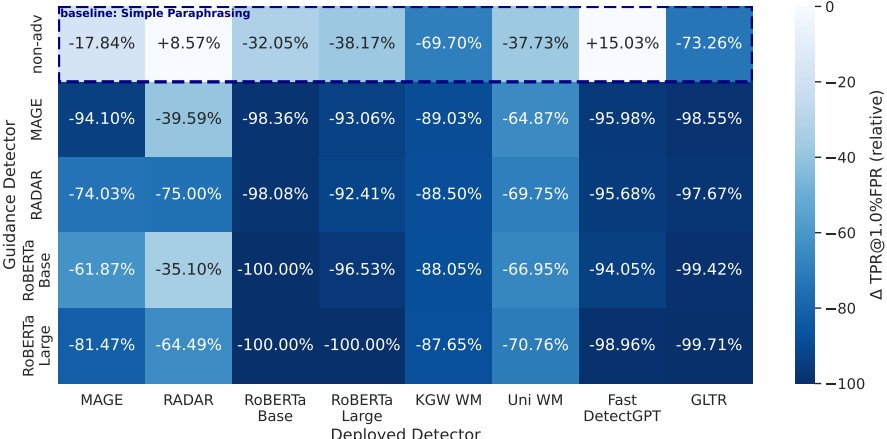

Figure 4: **Relative drop in T@1%F across all combinations of guidance and deployed detectors. The first row corresponds to simple (non-adversarial) paraphrasing baseline [15].** On average, simple paraphrasing leads to a 30.27% relative drop in T@1%F. In comparison, adversarial paraphrasing achieves significantly higher reductions—84.94% with MAGE as guidance, 86.89% with RADAR, 80.75% with OpenAI-RoBERTa-Base, and 87.88% with OpenAI-RoBERTa-Large. These results highlight both the universal effectiveness and transferability of our attack.

of the guidance detector, consistently lead to significant reductions in T@1%F compared to simple paraphrasing. This further underscores the universal effectiveness and transferability of our attack.

## 4.3 Efficiency of Adversarial Paraphrasing

Compared to simple paraphrasing, our method requires running a surrogate AI text detector at each decoding step by design. This naturally raises the question of how much latency this process introduces and whether it affects efficiency.

To evaluate this, we conducted adversarial paraphrasing on 100 randomly selected text samples, with five trials per configuration, using all four guidance detectors. To simplify measurement and report time on a per-sample basis (rather than per-batch), we used a batch size of 1 during these trials. In practical settings, including the main experiments described in Section 4.2, we employ larger batch sizes, which substantially reduce total runtime. The average paraphrasing time per sample across five trials is reported as the mean ± standard deviation (in seconds[3]) in Table 3. As shown, most guidance detectors introduce only minor latency compared to simple paraphrasing. The higher latency observed with MAGE arises from its LongFormer-based architecture, which requires longer inference time than the RoBERTa-based models.

| Method | Run time |
|---|---|
| Simple Paraphrase | $7.18 \pm 0.13$ |
| AdvPara (roblarge) | $10.20 \pm 0.18$ |
| AdvPara (robbase) | $8.64 \pm 0.11$ |
| AdvPara (mage) | $16.71 \pm 0.74$ |
| AdvPara (radar) | $9.69 \pm 0.20$ |

Table 3: Per-sample run time (in seconds) of adversarial paraphrasing over 100 randomly selected samples.

Overall, latency is primarily determined by the computational complexity of the guidance detector. From a computational cost perspective, the detector adds minimal overhead in terms of FLOPs relative to the paraphrasing LLM, which dominates total computation. In our experiments, the paraphrasing LLM contains approximately 8 billion parameters, whereas the detectors range from 100 to 350 million parameters—less than 5% of the paraphraser's size for the larger detectors and under 2% for the smaller ones.

## 4.4 Ablation Studies

Other than the guidance detector choice, the two main hyper-parameters related to our work are the $p$ and $k$ used in top-p and top-k masking. In the above, we have used $p = 0.99$ and $k = 50$, where $k = 50$ is also the default $k$ value set in HuggingFace Transformer library. To show how their settings

---

[3]Note that runtime per sample may vary with sequence length. To ensure a fair comparison—i.e., that AdvPara generates a similar number of tokens as simple paraphrasing—we report the mean and standard deviation of token counts in Appendix B.

affect our attack, we use OpenAI-RoBERTa-Large as the guidance detector and run ablation studies on the value of $p$ and $k$ on 500 randomly selected texts. Defaulting to $k = 50$ and $p = 0.99$ when varying the other, we report the **mean** AUC and TPR@1%FPR across all 8 deploy detectors in Table 4.

Fixing $k$ at 50, we can observe that when $p$ exceeds 0.99, the marginal gains in attack effectiveness diminish relative to the losses in text quality. Additionally, as one can imagine, further increasing slows down generation speed. Therefore, we select $p = 0.99$ as a balanced trade-off among attack effectiveness, text quality, and generation speed. With $p$ fixed at 0.99, the values of $k$ show relatively smaller influence, and mostly lead to comparable results. Hence we just stick with the default value in the HuggingFace Transformer library.

| Method | mean AUC | mean T@1%R | Rating |
|---|---|---|---|
| No Attack | 0.8419 | 0.4935 | – |
| Simple Paraphrase | 0.8588 | 0.2885 | $4.75 \pm 0.54$ |
| AdvPara (p=0.5) | 0.8606 | 0.2985 | $4.86 \pm 0.45$ |
| AdvPara (p=0.8) | 0.7895 | 0.2230 | $4.79 \pm 0.48$ |
| AdvPara (p=0.9) | 0.7309 | 0.1820 | $4.79 \pm 0.48$ |
| AdvPara (p=0.95) | 0.6816 | 0.1480 | $4.81 \pm 0.44$ |
| AdvPara (p=0.99) | 0.5433 | 0.0690 | $4.50 \pm 0.67$ |
| AdvPara (p=0.991) | 0.5381 | 0.0592 | $4.47 \pm 0.73$ |
| AdvPara (p=0.992) | 0.5244 | 0.0565 | $4.37 \pm 0.76$ |
| AdvPara (k=10) | 0.5596 | 0.0757 | $4.47 \pm 0.77$ |
| AdvPara (k=25) | 0.5448 | 0.0658 | $4.48 \pm 0.73$ |
| AdvPara (k=50) | 0.5433 | 0.0690 | $4.50 \pm 0.67$ |
| AdvPara (k=75) | 0.5421 | 0.0698 | $4.52 \pm 0.73$ |
| AdvPara (k=100) | 0.5426 | 0.0698 | $4.51 \pm 0.70$ |

Table 4: Attack effectiveness under different values of $p$ and $k$ used in top-p and top-k masking.

## 5 Quality Evaluation of Paraphrased Texts

We conduct a comprehensive evaluation to investigate the impact of adversarial paraphrasing on the perceived quality of AI-generated text, focusing on both semantic equivalence to the original text and clarity, fluency, and naturalness of the text itself. For this, we randomly sample 100 texts each from our three datasets (MAGE, KGW watermarked MAGE, and Unigram watermarked MAGE) and analyze them with four complementary studies: (1) perplexity scores (PPL), (2) SBERT [25] semantic similarity, (3) auto-rated quality from GPT-4o comparing paraphrased texts to their original AI counterparts, and (4) head-to-head win rates, also assessed by GPT-4o, comparing adversarial paraphrasing against simple paraphrasing. We provide representative examples of paraphrases in Table 1, with a much broader set of examples included in Appendix F to support qualitative manual inspection. Our findings highlight a nuanced trade-off between evading detectors and preserving textual quality.

| Text | PPL (mean±std) |
|---|---|
| Original AI | $14.94 \pm 10.40$ |
| Original Human | $15.02 \pm 7.71$ |
| Simple Paraphrase | $9.28 \pm 3.86$ |
| AdvPara (roblarge) | $14.26 \pm 4.97$ |
| AdvPara (robbase) | $14.86 \pm 6.32$ |
| AdvPara (mage) | $17.11 \pm 7.33$ |
| AdvPara (radar) | $14.26 \pm 5.13$ |

Table 5: Perplexity (PPL) scores for original AI-generated and human-written texts from the MAGE dataset, along with simple and adversarial paraphrased versions of the AI-generated texts.

**Perplexity Analysis.** We assess perplexity using LLaMA-3.1–8B-Instruct [21], comparing the original AI-generated text, simple paraphrases, and adversarial paraphrases. The results are summarized in Table 5. As shown in the table, human-written text typically exhibits higher perplexity than AI text, as human language tends to deviate more from the statistical regularities learned by LLMs. After applying simple paraphrasing, we observe a substantial improvement in the perplexity of AI text. This may be attributed to the fact that the model used for paraphrasing (LLaMA-3.1) is superior to the LLMs used for generating AI texts in the MAGE dataset (*e.g.* LLaMA). In contrast, adversarial paraphrasing yields perplexity scores that are comparable to the human texts from MAGE, which is reasonable given that our objective is to humanize AI texts.

| Method | SBERT Cos. Sim. |
|---|---|
| Simple Paraphrase | $0.8601 \pm 0.0880$ |
| AdvPara (roblarge) | $0.8082 \pm 0.1006$ |
| AdvPara (robbase) | $0.8128 \pm 0.0985$ |
| AdvPara (mage) | $0.8159 \pm 0.0982$ |
| AdvPara (radar) | $0.8095 \pm 0.1025$ |

Table 6: Cosine similarity of SBERT embeddings before and after paraphrasing.

**SBERT Similarity.** A common approach to assessing the semantic equivalence between two texts is to compute the cosine similarity between their SBERT embeddings [25]. Accordingly, we measure the cosine similarity of SBERT embeddings before and after paraphrasing, and compare adversarial paraphrasing with simple paraphrasing. The results are summarized in Table 6. Overall, although there is a slight reduction in the mean cosine similarity for adversarial paraphrasing, the values remain within an acceptable range given the high variance observed across samples.

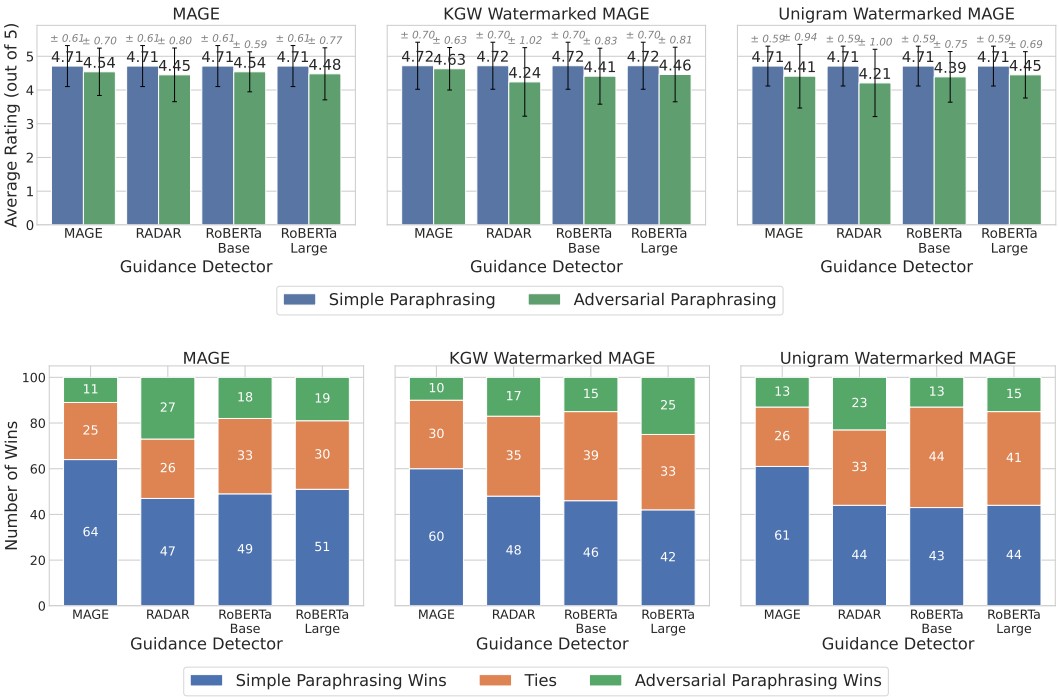

Figure 5: **GPT-4o automated text quality evaluations comparing simple and adversarial paraphrases.** The top row shows Likert-scale ratings for overall quality and semantic similarity to the original text. Though a slight trade off in text quality can be seen, the error bars show that the difference is not statistically significant. The bottom row presents head-to-head win rates, where in most cases, simple paraphrases outperform adversarial paraphrases less than half of the times.

**Auto-Rating with GPT-4o.** In order to simulate the human perception of text quality (i.e., alignment with the "gold standard" of fluency and coherence), we employ GPT-4o [23], as a judge LLM for automatic quality evaluations [32, 5, 13] with custom system and user prompts (see Appendix G). The judge model is tasked to rate the paraphrases when compared to their original corresponding AI text on a Likert scale of 1-5, in terms of quality and semantic similarity. The first row of Figure 5 shows the quality ratings for both baseline simple paraphrasing and adversarial paraphrasing. Though we observe a slight tradeoff in the text quality when compared to simple paraphrasing, in 87% of the times—averaged across all three datasets and four guidance detectors—adversarial paraphrases were rated 4 or 5 out of 5 (see Appendix H for detailed rating for each guidance detector and dataset). Note that the error bar in the figure shows that this difference between simple and adversarial paraphrasing is not statistically significant. While adversarial paraphrasing can lead to a higher perplexity score when compared to simple paraphrasing, our auto-rater study shows that both the paraphrases have a comparable text quality, making our attack a practical one.

**Win Rate Analysis.** To further compare the quality of simple and adversarial paraphrases, we conduct pair-wise evaluations using GPT-4o as a judge to compute their win rates [9]. Each pair consists of a simple and an adversarial paraphrase of an AI text, where the judge assigns a win, lose, or tie for the paraphrases. As shown in the second row of Figure 5, simple paraphrases win only less than half of the time in most cases. This finding reinforces the conclusion that adversarial paraphrasing can effectively evade detection with a slight tradeoff in text quality when compared to the prior simple pararaphrasing baseline.

# 6 Conclusion

With our comprehensive experiments, we demonstrate that our proposed **Adversarial Paraphrasing** is a universally transferable and effective attack for humanizing AI-generated text. Our text quality study shows that adversarial paraphrasing can drastically reduce detection rates with slight or no degradation in text quality majority of the time. Our findings underscore the vulnerability of existing detectors in the presence of a strong adversary. In the future, we believe our method can contribute to generating adversarial datasets for improving the robustness of trained detectors.

## Acknowledgment

This project was supported in part by a grant from an NSF CAREER AWARD 1942230, the ONR PECASE grant N00014-25-1-2378, ARO's Early Career Program Award 310902-00001, Army Grant No. W911NF2120076, the NSF award CCF2212458, NSF Award No. 2229885 (NSF Institute for Trustworthy AI in Law and Society, TRAILS), a MURI grant 14262683, DARPA AIQ grant HR00112590066, and an award from meta 314593-00001.

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

# A  Comparison with Watermark Stealing

In Section 4, we demonstrated the effectiveness of adversarial paraphrasing by comparing it against both simple and recursive paraphrasing. In this section, we extend our evaluation by comparing adversarial paraphrasing with the Watermark Stealing attack [12], a targeted approach specifically designed to compromise watermarking methods—unlike our more general-purpose (universal) attack.

Following the experimental setup proposed by Jovanović *et al.* [12], we use the LLaMA2-7B-Chat model [31], watermarked using the KGW scheme introduced by Kirchenbauer *et al.* [14], to generate a dataset of 2000 watermarked samples. Consistent with our prior experiments, the model is provided with the first 20 words of each of the 2000 AI-generated texts from the MAGE dataset [18] as a prefix and generates approximately 200 to 600 tokens per sample conditioned on that context. The watermarking parameters match those used in the original watermark stealing study [12]. Using the learned watermarking scheme, we then perform a scrubbing attack as described in the same study, employing both Mistral-7B [11] and LLaMA2-7B [31] as paraphrasers.

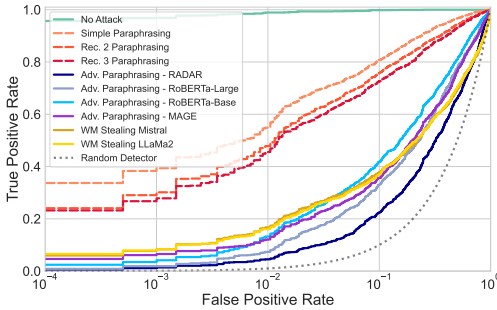

Figure 6: ROC curves illustrating the AI text detection performance of KGW watermark under different attacks, including simple paraphrasing, recursive paraphrasing, watermark stealing, and adversarial paraphrasing. The false positive rate (FPR) axes is displayed in log-scale to highlight fine-grained distinctions in the low-FPR region. It can be seen that adversarial paraphrasing outperforms all baselines, including watermark stealing, in pushing the detector's performance closer to that of a random one.

Figure 7: Text quality evaluations using GPT-4o, comparing watermark stealing and adversarial paraphrasing (guided by RADAR [8], which demonstrated the highest attack effectiveness in this case) against simple paraphrasing. The results show that adversarial paraphrasing produces higher-quality text than watermark stealing.

Figure 6 presents the ROC curves for KGW watermark detection performance under various attack scenarios, including simple paraphrasing, recursive paraphrasing, watermark stealing, and adversarial paraphrasing. From the ROC curves, it can be observed that while adversarial paraphrasing guided by certain detectors demonstrates slightly reduced effectiveness compared to watermark stealing in high false positive rate (FPR) regions, it consistently outperforms all baselines in degrading detector performance in the low FPR regime (FPR $\leq$ 1%). Notably, RADAR [8] proves to be the most effective guidance detector in this setting. Table 7 reports the same three key metrics introduced in our Experiments section: the Area Under the ROC Curve (AUC), the True Positive Rate at 1% False Positive Rate (T@1%F), and GPT-4o's automated quality ratings (Rating). The results

|  | KGW WM | | |
| --- | --- | --- | --- |
|  | AUC ($\downarrow$) | T@1%F ($\downarrow$) | Rating |
| No Attack | 0.999 | 0.989 | – |
| Simple Paraphrase | 0.930 | 0.549 | 4.93 $\pm$ 0.41 |
| Rec. Para. 2 | 0.903 | 0.478 | 4.95 $\pm$ 0.22 |
| Rec. Para. 3 | 0.889 | 0.455 | 4.92 $\pm$ 0.34 |
| WM Stealing (Mistral-7B) | 0.669 | 0.167 | 4.24 $\pm$ 0.99 |
| WM Stealing (LLaMA2-7B) | 0.670 | 0.161 | 4.28 $\pm$ 0.98 |
| AdvPara (RoBERTa-Large) | 0.703 | 0.073 | 4.62 $\pm$ 0.76 |
| AdvPara (RoBERTa-Base) | 0.751 | 0.132 | 4.76 $\pm$ 0.53 |
| AdvPara (MAGE) | 0.707 | 0.121 | 4.84 $\pm$ 0.46 |
| AdvPara (RADAR) | **0.619** | **0.045** | 4.40 $\pm$ 0.89 |

Table 7: Detection performance of KGW Watermark in distinguishing between AI-generated and human-written text under different attack scenarios. Metrics reported include AUC and TPR at 1% FPR. Additionally, we present the mean $\pm$ standard deviation of quality ratings given by GPT-4o. It can be observed that adversarial paraphrasing results in the lowest AUC and TPR@1%FPR after the attack.

indicate that adversarial paraphrasing guided by RADAR yields the lowest AUC and T@1%F values post-attack. Figure 7 further details the text quality assessments for the watermark stealing and adversarial paraphrasing attacks, compared against simple paraphrasing. It shows that adversarial paraphrasing results in better text quality compared to watermark stealing.

# B    Text Token Counts in Efficiency Assessment

The exact runtime per sample may vary depending on the sequence length, which is inherently determined by the length of the input text. To ensure a fair comparison—such that adversarial paraphrasing generates a comparable number of tokens to simple paraphrasing—we report the mean and standard deviation of token counts in Table 8. As shown, both simple paraphrasing and adversarial paraphrasing with different guidance detectors produce a similar mean number of tokens. However, adversarial paraphrasing exhibits a larger variance in token counts, reflecting greater variability in output length.

| Text | Mean Token Count | Std Dev |
|---|---|---|
| Original texts | 173.73 | 38.46 |
| Simple Paraphrase | 170.81 | 39.69 |
| AdvPara (RoBERTa-Base) | 175.30 | 51.27 |
| AdvPara (RoBERTa-Large) | 171.25 | 45.39 |
| AdvPara (RADAR) | 169.68 | 60.07 |
| AdvPara (MAGE) | 164.18 | 54.39 |

Table 8: Mean and standard deviation of token counts for the original, simple paraphrased, and adversarially paraphrased texts used in our efficiency evaluation. All token counts were obtained using the LLaMA-3 tokenizer.

# C    Detailed Token Statistics for Evaluated Datasets

We report the detailed token statistics for all datasets used in our main experiments in Table 9. The token counts are obtained from the LLaMA-3 tokenizer.

| Dataset | Min # tokens | Max # tokens | Mean # tokens |
|---|---|---|---|
| MAGE human texts | 110 | 305 | $\sim$179 |
| MAGE AI texts | 110 | 525 | $\sim$175 |
| KGW watermarked texts | 199 | 602 | $\sim$269 |
| Unigram watermarked texts | 161 | 602 | $\sim$319 |

Table 9: Token statistics for the texts used in our evaluation, obtained from the LLaMA-3 tokenizer.

# D    What Happens if Detector Guidance is Not Applied at All Decoding Steps?

In our default configuration, detector guidance is applied at every iteration of adversarial paraphrasing (Figure 1). This setup achieves high attack effectiveness (Section 4.2) while maintaining acceptable latency (Section 4.3). In this section, we conduct an ablation study to investigate the impact of applying detector guidance less frequently. Specifically, guidance is applied once every $N$ steps during the auto-regressive generation process of the paraphraser. In this setting, the paraphraser generates $N-1$ tokens in a standard (non-adversarial) manner, and the $N$-th token is then sampled with detector guidance. For the guidance detector, we employ OpenAI-RoBERTa-Large and perform adversarial paraphrasing on 500 randomly selected text samples. The resulting mean AUC and TPR@1%FPR across all eight deployed detectors are reported in Table 10.

It can be observed that applying detector guidance at every decoding step yields the highest attack effectiveness, while performance gradually decreases as $N$ increases. Nevertheless, even with $N=5$, adversarial paraphrasing maintains superior attack effectiveness compared to the simple paraphrasing baseline. Moreover, using a larger $N$ reduces attack latency, as fewer detector-guided steps are required.

| Method | Mean AUC | Mean T@1%FPR |
|---|---|---|
| No Attack | 0.8418 | 0.4935 |
| Simple Paraphrase | 0.8588 | 0.2885 |
| AdvPara (RoBERTa-Large) (N=1) | **0.5500** | **0.0740** |
| AdvPara (RoBERTa-Large) (N=2) | 0.6632 | 0.1333 |
| AdvPara (RoBERTa-Large) (N=3) | 0.7266 | 0.1690 |
| AdvPara (RoBERTa-Large) (N=4) | 0.7593 | 0.1820 |
| AdvPara (RoBERTa-Large) (N=5) | 0.7788 | 0.2115 |

Table 10: Ablation on detector guidance frequency ($N$) during adversarial paraphrasing. Results are averaged over 8 deployed detectors. Lower AUC and T@1%FPR indicate stronger attack effectiveness.

## E   Failure Cases and Backfires

While our attack is highly effective, it is not always successful on every single sample, and occasional backfiring can occur—though such instances are rare. Assuming a fixed detection threshold of 0.5 for NN-based detectors and focusing on the more challenging *transfer* setting (i.e., different guidance–detector pairs), we analyzed a total of 24,000 samples.

Out of these, 5,742 samples exhibited an *increase* in detection score after paraphrasing. Among them:

- **1,773** examples were already flagged as AI-generated before paraphrasing. In these cases, the attack simply failed to help.

- **2,974** examples were classified as human both before and after paraphrasing, meaning they consistently evaded detection—thus, the attack did not worsen the outcome.

- **995** examples were originally classified as human but crossed the 0.5 threshold after paraphrasing, resulting in a *backfire* where the paraphrased text was more likely to be detected as AI-generated.

In summary:

- The attack **failed** in 1,773 cases (~7.4% of all samples).

- It **backfired** in 995 cases (~4.15% of all samples).

Despite these occasional failures, our method consistently outperforms baselines in reducing AUC and T@1%F, as demonstrated in our results. Such failures are expected in adversarial settings but do not undermine the overall robustness, effectiveness, or transferability of our approach.

## F   More Examples of Paraphrased Texts

Tables 12 through 18 present additional examples of original AI-generated texts along with their corresponding simple and adversarial paraphrases, the latter guided by OpenAI-RoBERTa-Large [30].

Tables 12 to 15 include examples in which both simple and adversarial paraphrases received a rating of 5. Table 16 shows instances where the simple paraphrases received a rating of 5, while the adversarial paraphrases received a rating of 4. For these cases, we also include the justification provided by GPT-4o for the score assigned. Tables 17 and 18 presents examples where neither the simple nor the adversarial paraphrases achieved a rating of 5. Justifications from GPT-4o are again included for these examples.

Across the 100 simple paraphrased texts, 76 received a rating of 5, 17 received a rating of 4, and 7 received a rating of 3. For the 100 adversarially paraphrased texts, 64 received a rating of 5, 21 received a rating of 4, 14 received a rating of 3, and 1 received a rating of 2. This shows that simple and adversarial paraphrases result in comparable text quality.

# G System and User Prompt for GPT-4o Quality Rating and Win-Rate Judgment

We configure GPT-4o as a text quality rater using the system prompt shown in Figure 8, and as a win rate judge using the system prompt shown in Figure 9. The user prompt for quality rating is shown in Figure 10, and the user prompt for win rate judging is shown in Figure 11.

---

You are an expert linguist and paraphrase evaluator. Your task is to assess the quality of a paraphrased text compared to the original source text. Use the following scoring criteria:

5 - Approximately equivalent: Meaning is preserved; differences are only in wording or structure.
4 - Nearly equivalent: Meaning is mostly preserved; minor factual details differ.
3 - Somewhat equivalent: Some meaning is preserved; important details or meanings differ.
2 - Topically related: The texts are on the same topic but most meaning is lost.
1 - Not topically related: The texts are not related in topic or meaning.

Provide your final output as a JSON object in this format:
{
"score": <score from 1 to 5>,
"justification": "<brief explanation>"
}

---

Figure 8: The system prompt used to configure GPT-4o as a quality rater for evaluating the quality of paraphrased texts.

---

You are a neutral and highly capable evaluator. Your task is to compare two versions of a text and select the one that communicates the ideas more clearly, accurately, and fluently. Consider:

- Fidelity of meaning (how clearly the core message is conveyed)
- Clarity and conciseness
- Grammatical correctness and fluency
- Naturalness and appropriateness of phrasing

Give your vote solely based on quality. If one of them is a CLEAR winner, ONLY then vote for that one. Otherwise, vote for 'tie'.

Respond with **only one of the following**, and nothing else:
- text1
- text2
- tie

---

Figure 9: The system prompt used to configure GPT-4o as a win-rate judge for comparing the quality of simple and adversarial paraphrases.

---

Evaluate the following paraphrase using the criteria above:

Original Text:
<original_text>

Paraphrased Text:
<paraphrased_text>

What score (1 to 5) would you assign to this paraphrase, and why?

---

Figure 10: The user prompt for querying GPT-4o for the quality ratings.

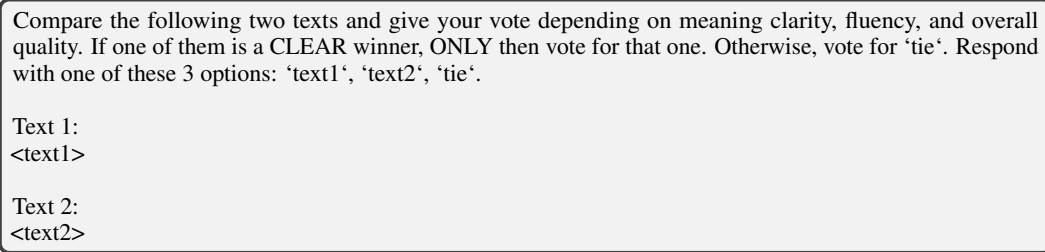

Figure 11: The user prompt for querying GPT-4o for win rate judging.

## H Detailed GPT-4o Ratings for All Detectors and Datasets

In Table 11, we report the detailed GPT-4o quality ratings (mean $\pm$ std) and the percentage of high-quality scores (ratings 4 and 5) for all paraphrased outputs across the three datasets involved. It can be observed that while adversarial paraphrasing leads to a slightly lower average quality rating compared to simple paraphrasing, the difference is not statistically significant. On average, across all datasets and guidance detectors, 87% of the adversarially paraphrased texts received a quality rating of 4 or 5.

| Original Text | Guidance Detector | Avg. Rating (mean $\pm$ std) | | Rating 5&4 (in %) | |
|---|---|---|---|---|---|
| | | Simple Para. | AdvPara | Simple Para. | AdvPara |
| MAGE | mage | 4.71 $\pm$ 0.61 | 4.54 $\pm$ 0.70 | 92% | 88% |
| | radar | | 4.45 $\pm$ 0.80 | | 85% |
| | robbase | | 4.54 $\pm$ 0.59 | | 95% |
| | roblarge | | 4.48 $\pm$ 0.77 | | 85% |
| KGW Watermarked MAGE | mage | 4.72 $\pm$ 0.70 | 4.63 $\pm$ 0.63 | 92% | 94% |
| | radar | | 4.24 $\pm$ 1.02 | | 79% |
| | robbase | | 4.41 $\pm$ 0.83 | | 86% |
| | roblarge | | 4.46 $\pm$ 0.81 | | 86% |
| Unigram Watermarked MAGE | mage | 4.71 $\pm$ 0.59 | 4.41 $\pm$ 0.94 | 95% | 85% |
| | radar | | 4.21 $\pm$ 1.00 | | 82% |
| | robbase | | 4.39 $\pm$ 0.75 | | 86% |
| | roblarge | | 4.71 $\pm$ 4.45 | | 93% |

Table 11: GPT-4o quality ratings (mean $\pm$ std) and percentage of high-quality scores (ratings 4 and 5) for outputs rewritten using different guidance detectors. Ratings are split across simple and adversarial paraphrase scenarios.

## I Computation Resources

We utilize two NVIDIA RTX A6000 GPUs to host both the paraphraser language model and the guidance AI text detector. Notably, since our approach is compatible with any instruction-tuned language model for paraphrasing and any neural network–based AI text detector, the computational requirements may vary depending on the specific models used.


| Text | Rating |
|---|---|
| **Original AI Text.** Automatic software plagiarism detection tools are widely used in educational settings to ensure that submitted work was not copied. These tools have grown in use together with the rise in online learning and students frequently submit papers electronically using these services, which automatically analyze content for similarities between their own creations and those of other users. While this approach does identify some instances where a service has detected unauthorised usage of its products, it can miss many cases - especially when there is no clear pattern or similarity across different sources. To overcome the problem of false positives for both low-quality original content as well as plagiarized content from third parties, today's research team led by Scott Pfeiffer at MIT created an automated system that predicts whether plagiarism is likely based on several factors including: how similar two web pages appear; what text they share; how easily their content resembles each other; if neither source appears to be a trusted authority; and the strength of a paper's author citation record. The results suggest that plagiarism may occur more often than previously thought. | – |
| **Simple Paraphrase.** In the era of digital learning, the widespread use of educational software has led to a surge in the adoption of plagiarism detection tools to verify the originality of student submissions. These automated services have become increasingly prevalent, particularly with the rise of online learning, where students frequently submit electronic assignments. While these tools have been effective in identifying some instances of unauthorized use, they are not foolproof and may overlook many cases, especially when there is no apparent pattern or similarity between sources. To address the issue of false positives for both low-quality original content and plagiarized work from third parties, a team of researchers at MIT, led by Scott Pfeiffer, have developed an innovative system that predicts the likelihood of plagiarism based on several key factors, including the visual similarity of web pages, shared text, content resemblance, the credibility of sources, and the strength of an author's citation record. The findings suggest that plagiarism may be more widespread than previously assumed. | 5 |
| **Adversarial Paraphrase.** As online education expands, plagiarism detection software has become an essential tool to ensure original work is being submitted by students. With more assignments submitted electronically, software analyzes the content for similarities, flagging suspicious matches with existing sources. However, relying solely on similarity checks can be incomplete, missing cases without distinctive patterns or duplicates. Researchers led by Scott Pfeiffer at MIT have designed an innovative system that goes beyond this approach. This system takes into account a range of factors, such as web page visual similarity, shared content, and content resemblance, to predict whether plagiarism has occurred. It also considers the trustworthiness of sources and authors' citation records. According to the study, plagiarism might be more prevalent than initially thought, highlighting the need for advanced methods to tackle this issue effectively. | 5 |
| **Original AI Text.** It was the day of the Mardi Gras float parade and all of the children in the neighborhood were excited to see the colorful floats and costumes. But one little girl, Sadie, was not feeling well. She had chicken pox and was stuck at home while everyone else was out having fun. Sadie's mom tried to make her feel better by putting on a Mardi Gras mask and giving her a beaded necklace, but it was just not the same as being there. Sadie lay in her bed, watching the parade go by, when she suddenly had an idea. She got out of bed and went to the window. Her mom had left the ladder out so she could reach the top of the float. Sadie climbed up and found herself right in the middle of the parade! She waved to all of her friends as they passed by, and they were so surprised to see her. It was the best Mardi Gras ever, even though she did have to miss the parade itself. | – |
| **Simple Paraphrase.** As the vibrant Mardi Gras parade unfolded in the neighborhood, the children's excitement was palpable. But amidst the revelry, one young girl, Sadie, was confined to her home, her enthusiasm dampened by a bout of chicken pox. Her mother attempted to lift her spirits by donning a Mardi Gras mask and bestowing a beaded necklace, yet the experience seemed hollow without the actual parade. Sadie's gaze drifted out the window, where she spotted the colorful floats gliding by, and inspiration struck. She sprang from her bed, seized the opportunity presented by the ladder left behind by her mother, and climbed up to the window sill. As she reached the top, Sadie found herself in the midst of the parade, surrounded by the festive atmosphere she had been missing. With a beaming smile, she waved to her friends as they passed, their faces aglow with surprise. Despite missing the parade itself, Sadie's impromptu adventure turned out to be the most unforgettable Mardi Gras experience of all. | 5 |
| **Adversarial Paraphrase.** As Mardi Gras floats rolled down the streets, excitement radiated through the neighborhood, infecting children with its contagious energy. Except for one little girl, Sadie, who lay bedridden with chicken pox, watching the colorful spectacle from afar. Her mom endeavored to console her, adorning Sadie with a festive mask and a beaded necklace, but the youngster yearned for more. It wasn't until a brainstorm struck Sadie that her confinement transformed into an impromptu adventure. Seizing the moment, she quietly climbed the ladder outside and discovered herself at the very center of the Mardi Gras parade! Overcome with glee, she waved merrily to friends and acquaintances, their stunned faces etched with surprise. It turned out to be an unforgettable Mardi Gras experience, even if she'd had to forgo the initial procession itself. | 5 |
| **Original AI Text.** In 2000, ABC launched a web-based promotional campaign focused around its circle logo, also called "the dot," in which comic book character Little Dot prompted visitors to "download the dot," which was a free screensaver. The promotional campaign utilized both online and offline elements, including television ads and an interactive website featuring a game called "Dot Dodge." The on-air spots featured Little Dot, a 3-D animated character that urged viewers to download the dot from the ABC website. The site also offered the ability to customize the dot with photos, quotes, and music. Additionally, users could play the Dot Dodge game in which they had to dodge obstacles as they moved around a maze. The campaign was successful in driving significant web traffic to ABC's website, as well as increasing positive brand awareness and engagement. It also won several awards, including a Golden Beacon Award for Best Use of Interactive Technology and an Outstanding Achievement Award from the International Academy of Digital Arts Sciences. | – |
| **Simple Paraphrase.** 2000 marked a pivotal year for ABC, as they launched a groundbreaking promotional campaign centred around their iconic circle logo, affectionately dubbed "the dot". The initiative involved a multifaceted approach, combining online and offline elements to create a memorable experience for audiences. The campaign's centrepiece was a 3D animated character, Little Dot, who encouraged viewers to download the dot, a free screensaver, from the ABC website. The interactive site also allowed users to personalize the dot with their preferred images, quotes, and music. Furthermore, the campaign featured a thrilling game, Dot Dodge, where players navigated a maze while avoiding obstacles. The campaign's success was evident in the significant surge in website traffic, enhanced brand recognition, and increased engagement. It also garnered widespread acclaim, earning prestigious awards such as the Golden Beacon Award for Best Use of Interactive Technology and the Outstanding Achievement Award from the International Academy of Digital Arts Sciences. | 5 |
| **Adversarial Paraphrase.** In 2000, ABC rolled out an immersive campaign centred on its iconic circular logo, affectionately known as 'The Dot'. At the heart of this initiative was Little Dot, a cartoon character encouraging visitors to 'collect The Dot', a complimentary desktop screensaver. Leveraging both digital and offline strategies, the campaign spanned TV commercials, an interactive website featuring the addictive game 'Dot Dodge', and offline elements. These eye-catching TV spots introduced viewers to Little Dot, a three-dimensional animated protagonist persuading them to download The Dot from ABC's website. Visitors could personalise The Dot by uploading photos, quotes and music. The site also allowed users to indulge in Dot Dodge, navigating through a challenging obstacle course. The campaign generated impressive web traffic for ABC's website, accompanied by enhanced brand recognition and engagement. Notably, it won accolades like the Golden Beacon Award for Best Use of Interactive Technology and the International Academy of Digital Arts Sciences' Outstanding Achievement Award. | 5 |

Table 12: Examples of original AI texts with their simple and adversarial paraphrases (guided by OpenAI-RoBERTa-Large [30]). GPT-4o quality ratings are provided for each paraphrased version.

| Text | Rating |
|---|---|
| **Original AI Text.** If addiction is a disease, why do most addictions end spontaneously, without treatment? Why did some 75 of heroin-addicted Vietnam vets kick the drug when they returned home? It's hard to say exactly why some addictions end spontaneously without treatment. It is possible that the person was motivated to change their behavior and developed the necessary coping skills to do so. It could also be due to other factors such as external support from friends and family, positive reinforcement from professionals, or simply a change in environment. The Vietnam vets may have had a strong desire to leave the drug behind and return to their normal lives. Additionally, they likely faced a much different environment upon returning home which provided them with fewer opportunities to use the drug. | – |
| **Simple Paraphrase.** The puzzling phenomenon of spontaneous recovery from addiction has left many wondering why most addictions tend to resolve on their own without professional intervention. For instance, it's striking that nearly three-quarters of heroin-addicted Vietnam veterans were able to overcome their dependence on the drug upon returning home. While it's challenging to pinpoint the exact reasons behind this phenomenon, several factors might have contributed to their success. Perhaps the individuals in question were driven to reform their habits and developed the necessary skills to do so. Alternatively, external support from loved ones, positive reinforcement from professionals, or a change in environment could have also played a role. In the case of the Vietnam vets, their strong desire to reintegrate into civilian life and leave their addiction behind might have been a powerful motivator. Furthermore, the vastly different environment they returned to, which offered fewer opportunities to use the drug, may have also been a significant factor in their ability to overcome their addiction. | 5 |
| **Adversarial Paraphrase.** Despite addiction being likened to a disease, a puzzling phenomenon exists where many addictions naturally dissipate without treatment. Take, for instance, the high proportion of Vietnam veterans addicted to heroin who miraculously kicked the habit upon returning home. There is no clear explanation as to why this happens. Perhaps individuals simply resolved to change their behavior, cultivating the necessary life skills to overcome addiction. It's also plausible that external influences played a role, such as the emotional support of loved ones, motivational reinforcement from medical professionals, or an altered environment. It's possible that the vets simply longed to distance themselves from the drug and reclaim their pre-war way of life. The sudden change in environment upon returning home also limited their opportunities to continue using the drug. | 5 |
| **Original AI Text.** It's funny. Being a hitman is nothing like what you see in the movies. No awesome ghillie suit, no barcode tattoo on your bald-ass head. You just have to know how to shoot and kill people." He said he doesn't want his new show "to be about being an assassin" but rather showing that it could happen today. But when it comes to killing someone who is so powerful, it still has its challenges. Lundgren said: "You can never prepare for something as big as this. It takes everybody with the same mindset - the same commitment and dedication to get through it or survive it because it's going to end differently than anybody expects." | – |
| **Simple Paraphrase.** It's a far cry from the cinematic portrayals of hitmen. The reality is more about honed skills and precision, rather than dramatic getups and tattoos. The conversation turned to his latest show, which he emphasized is not about being an assassin, but rather a realistic portrayal of what could happen in today's world. However, even in the face of taking down a powerful target, there are significant challenges to overcome. Lundgren noted that "you can't prepare for something of this magnitude. It demands a united front, a shared commitment and dedication to overcome the obstacles and emerge unscathed, as the outcome is always unpredictable." | 5 |
| **Adversarial Paraphrase.** Contrary to cinematic portrayals, the reality of being a hitman is surprisingly unglamorous. Forget elaborate disguises or conspicuous tattoos; success hinges on proficient marksmanship and the capacity for cold-bloodedness. Lundgren candidly reveals his latest project aims not to glamorize assassination but rather to humanize the very possibility that it could unfold in today's world. However, he concedes that even when dealing with high-profile targets, formidable challenges still arise. "Nothing can fully prepare you for the scope and complexity of such a massive undertaking," he says, "It demands unwavering unity of purpose and unwavering commitment among all parties involved, lest it unfold in ways no one can predict, leaving few with any sense of closure." | 5 |
| **Original AI Text.** Various microRNAs (miRNAs) that counteract viruses present in hosts can decrease the replication of the simian immunodeficiency virus (SIV). Several research studies have shown that certain miRNAs, namely miR-28, miR-125b, miR-150, and miR-223, possess the potential to inhibit the replication of SIV. The degradation of SIV viral RNA or the inhibition of its translation, which reduces viral replication, is caused by the binding of specific targets in the 3' untranslated region (3'UTR) of SIV by four miRNAs. The interaction between SIV and its host is not simple, as other cellular factors and pathways play a role in controlling viral replication that should be considered. Additionally, further investigation is required to examine the involvement of these miRNAs in the replication of HIV-1 (human immunodeficiency virus type 1) that has a strong similarity to SIV. To sum up, although miR-28, miR-125b, miR-150, and miR-223 are capable of reducing SIV replication through direct means, the regulation of the virus is a complicated process that involves a variety of host and viral factors. | – |
| **Simple Paraphrase.** Research has revealed that several microRNAs, including miR-28, miR-125b, miR-150, and miR-223, possess the ability to impede the replication of the simian immunodeficiency virus (SIV) by targeting specific sequences in the 3' untranslated region of the virus. These miRNAs can either degrade SIV viral RNA or inhibit its translation, thereby diminishing viral replication. However, the interplay between SIV and its host is complex, involving various cellular factors and pathways that modulate viral replication. Furthermore, the role of these miRNAs in the replication of human immunodeficiency virus type 1 (HIV-1), which shares significant similarities with SIV, warrants further investigation. In summary, while miR-28, miR-125b, miR-150, and miR-223 can directly restrict SIV replication, the regulation of the virus is a multifaceted process influenced by both host and viral factors. | 5 |
| **Adversarial Paraphrase.** Numerous miRNAs have been discovered that counteract viruses infecting host cells, leading to the diminution of the replication of the simian immunodeficiency virus (SIV). Investigations have pinpointed several miRNAs - namely, miR-28, miR-125b, miR-150, and miR-223 - which exhibit an inhibitory effect on SIV replication. Through binding to specific targets within the 3' untranslated region of SIV RNA, these miRNAs orchestrate the degradation or inhibition of SIV translation, subsequently curtailing viral replication. Notably, the intricate interaction between SIV and its host necessitates the consideration of multiple cellular factors and pathways governing viral replication. Furthermore, comprehensive analysis is essential to determine the extent of the role these miRNAs play in the replication of human immunodeficiency virus type 1 (HIV-1), given the similarities it shares with SIV. Ultimately, while miR-28, miR-125b, miR-150, and miR-223 effectively curb SIV replication directly, the regulation of the virus remains a complex process entailing multiple host and viral factors. | 5 |

Table 13: Examples of original AI texts with their simple and adversarial paraphrases (guided by OpenAI-RoBERTa-Large [30]). GPT-4o quality ratings are provided for each paraphrased version.

| Text | Rating |
|---|---|
| **Original AI Text.** I hope there was a good reason for Steve to flake on his appointment for window tinting. Attempted to contact him multiple times with no answer and no return call. I'm sorry to hear that Steve missed his appointment without any notification. Perhaps there was an emergency or something important came up that he had to deal with. However, it would have been courteous of him to inform you beforehand. If you still have his contact information, you may try reaching out to him one more time to see if he would like to reschedule or if there was anything you could do to help. | – |
| **Simple Paraphrase.** It's disheartening to find out that Steve cancelled his window tinting appointment without any prior notice or explanation. Despite making repeated attempts to reach out to him, I was met with silence. While I'm willing to give him the benefit of the doubt, assuming an unforeseen circumstance might have arisen, I do think it would have been considerate of him to communicate his change of plans earlier. If you still have his contact details, it may be worth one more attempt to connect with him and explore the possibility of rescheduling or offering assistance if needed. | 5 |
| **Adversarial Paraphrase.** Regrettably, it seems that Steve failed to honor his commitment for window tinting. Despite making repeated efforts to reach him, my calls went unanswered and I received no callback. It's disheartening to think that he skipped his scheduled appointment without prior notification. Although unforeseen circumstances might have arisen, I do think it would have been polite for him to notify me ahead of time. If you still have his contact details, I would suggest attempting to re-establish contact one final time to either reschedule the appointment or see if there's any assistance I can offer him in his time of need. | 5 |
| **Original AI Text.** The Waking the Dragon tower was granted permission by Wrexham council back in February 2011. Businessman Simon Wingett is trying to raise PS2.5m to build the bronze sculpture, cultural centre and entertainment complex on a site next year's Great North Run finish line at Llangollen. With three days of bidding remaining for Britain's most prestigious art prize - Art Fund Prize 2013 - it has been announced that the winner will receive 100,000 plus an award worth around 15,000 from Arts Council England (ACE). The runners up each receive prizes of between 7,500 and 10,000 from ACE. In total, the awards for all nominated works are worth over 450,000. | – |
| **Simple Paraphrase.** In a significant development, the Waking the Dragon tower project received the green light from Wrexham council in February 2011. Entrepreneur Simon Wingett is now working to secure a substantial PS2.5 million investment to construct a bronze sculpture, cultural hub, and entertainment complex on the site of next year's Great North Run finish line in Llangollen. With only a few days remaining to submit bids for the prestigious Art Fund Prize 2013, it has been revealed that the winner will receive a staggering 100,000-pound prize, accompanied by an additional 15,000-pound award from Arts Council England. The runners-up will receive prizes ranging from 7,500 to 10,000 pounds from ACE, totaling over 450,000 pounds in awards for all nominated works. | 5 |
| **Adversarial Paraphrase.** In 2011, Wrexham council greenlit the ambitious project known as the Waking the Dragon tower. Developer Simon Wingett aims to raise PS2.5m to construct a striking bronze sculpture, a multifaceted cultural hub, and entertainment destination at the iconic location marking the finish line of the Llangollen Great North Run next year. Meanwhile, the countdown has begun for the 2013 Art Fund Prize, with just days to go before the winner is announced. The esteemed accolade, valued at PS100,000, is complemented by an additional prize worth around 15,000 from Arts Council England (ACE). Runners-up will receive awards ranging from PS7,500 to PS10,000, totalling a combined value of over PS450,000 for all nominated works. | 5 |
| **Original AI Text.** One of the challenges in large-scale information retrieval (IR) is to develop fine-grained and domain-specific methods to answer natural language questions. Despite the availability of numerous sources and datasets for domain-specific IR (DS-IR), most DS-IR techniques rely on supervised machine learning techniques. Therefore, approaches that are capable of utilizing unlabeled data in DS-IR can be of particular interest. To that end, we propose a new DS-IR approach that exploits the power of unsupervised learning to capture domain-specific patterns and structures. The proposed framework consists of three main components: an unsupervised learning algorithm to automatically discover the hidden patterns and structures from the target DS-IR domain, a model to incorporate these structures to the IR process, and a scoring mechanism for the retrieved documents. We present experiments on two different DS-IR domains: biomedical and patents. We report results for two different evaluation measures, which indicate the effectiveness of the proposed approach. | – |
| **Simple Paraphrase.** In the realm of large-scale information retrieval, a crucial challenge lies in developing techniques that can accurately address complex, domain-specific queries. Despite the abundance of datasets and sources available for domain-specific IR, most existing methods rely on supervised machine learning approaches. However, exploring the potential of unsupervised learning techniques in this domain can be a valuable direction. To address this gap, we introduce a novel DS-IR approach that harnesses the power of unsupervised learning to uncover domain-specific patterns and relationships. Our framework comprises three key components: a pattern discovery module that automatically identifies hidden structures in the target domain, a structural integration module that incorporates these patterns into the IR process, and a scoring mechanism that evaluates the retrieved documents. We demonstrate the effectiveness of our approach through experiments on two distinct DS-IR domains: biomedical and patents, using two evaluation metrics. | 5 |
| **Adversarial Paraphrase.** Harnessing the potential of unsupervised machine learning is a critical challenge in the large-scale information retrieval domain, where fine-tuned, domain-specific solutions are desperately needed to effectively answer complex natural language queries. Despite the wealth of readily available datasets for domain-specific IR, many prevailing approaches still rely heavily on supervised machine learning methodologies. Recognizing the untapped potential of unlabeled data, our novel domain-specific IR technique leverages the power of unsupervised learning to distill unique patterns and structures peculiar to specific domains. Comprising three key components - a unsupervised algorithm to automatically unearth these domain-specific patterns, a framework to integrate them seamlessly into the retrieval process, and a scoring mechanism to gauge retrieved document relevance - our approach promises to revolutionize the DS-IR landscape. Experimental evaluations across two distinct domains, biomedical and patents, validate the effectiveness of our proposed technique through robust performance on various evaluation metrics. | 5 |

Table 14: Examples of original AI texts with their simple and adversarial paraphrases (guided by OpenAI-RoBERTa-Large [30]). GPT-4o quality ratings are provided for each paraphrased version.

| Text | Rating |
|---|---|
| **Original AI Text.** The United Methodist Church is organized into conferences. The highest level is called the General Conference and is the only organization which may speak officially for the church. The General Conference also includes all of us who have been engaged in ministry or are involved in ministry, whether we know it or not. Our Mission To serve as a spiritual church that is grounded with values and purpose. We believe that every individual has the right to participate in God's plan for His worshiping people through a Divine Word; we strive to inspire not only those who listen to our prayers but those who go out and pray with their hearts as well. | – |
| **Simple Paraphrase.** The United Methodist Church is structured into conferences, with the General Conference serving as the supreme governing body that represents the church's official voice. This assembly encompasses all individuals who have dedicated themselves to ministry, whether actively involved or not. Our fundamental purpose is to function as a spiritual community rooted in values and direction. We uphold the notion that every individual has the privilege to participate in God's plan for His people, as revealed through the sacred text; our aim is to not only move those who hear our prayers but also to empower those who respond with heartfelt devotion. | 5 |
| **Adversarial Paraphrase.** Within the United Methodist Church, the organisational structure revolves around conferences, with the General Conference standing as the highest echelon and the only authoritative voice of the denomination. Beyond its members, the General Conference encompasses a vast, invisible community encompassing everyone touched by the church, regardless of their awareness or direct involvement. At its core, our mission is to be a spiritually nourishing church built upon firm values and purpose. We firmly believe that each individual possesses a vital role to play in the grand plan for the collective worship of God, facilitated through the Holy Word; accordingly, our mission extends not only to those who attentively listen to our prayers but also to those who pray with unwavering dedication in their hearts. | 5 |
| **Original AI Text.** DiDi's co-founder and president Jean Liu told close associates that she plans to step down as she expects the government to eventually take over the company and appoint new management. Photo: Bloomberg Didi Chuxing co-founder to step down from management Jean Liu, co-founder and president of China's ride-hailing giant Didi Chuxing, plans to step down from management and will focus on the company's strategy and business development, Reuters reports, citing people with knowledge of the matter. Didi has been beset by a series of high-profile scandals, including the murders of two of its customers. It has since made major changes to its ride-hailing business and rolled out a string of safety measures. Didi's former president, Liu Qing, also left the company last year after one of the murders. Liu, who has been with Didi since 2014, told close associates that she plans to step down as she expects the government to eventually take over the company and appoint new management, the report said. Didi's co-founder, Cheng Wei, currently serves as chief executive. | – |
| **Simple Paraphrase.** According to sources close to the matter, Jean Liu, co-founder and president of Didi Chuxing, is poised to relinquish her management role, with the expectation that the government will assume control of the company and install a new leadership team. This decision comes amid a tumultuous period for the Chinese ride-hailing giant, which has faced a series of high-profile controversies, including the tragic deaths of two customers. In an effort to restore public trust, Didi has implemented a range of safety measures and revamped its ride-hailing business model. The company's former president, Liu Qing, departed last year following one of the murders. Liu, who has been with Didi since 2014, reportedly plans to focus on strategy and business development, leaving the reins to Cheng Wei, the company's co-founder and current chief executive. | 5 |
| **Adversarial Paraphrase.** According to credible sources, DiDi Chuxing co-founder and president Jean Liu intends to relinquish her executive responsibilities and devote more time to charting the company's strategic direction and exploring business opportunities. The move follows a series of high-profile controversies, including two tragic incidents that resulted in fatalities, as well as significant overhauls and safety measures implemented to revamp the ride-hailing service. Interestingly, Liu's decision coincides with her own personal transition out of the executive ranks, echoing a similar path taken by her predecessor, Liu Qing, who parted ways with DiDi after a deadly incident last year. Interestingly, Liu Qing had also left her role as president following a fatality connected to DiDi. Jean Liu's intention to step down, sources close to the matter claim, is premised on a tacit understanding that the Chinese government will assume control and appoint new management. | 5 |
| **Original AI Text.** Susan had always wanted to learn how to play tennis. She was excited when she saw a sign for a class being offered at the local park. She signed up and was ready to start learning. On her first day, she met her instructor who showed her the basics of playing tennis. He taught her about proper form, swinging the racket, and how to keep score. Susan was a quick learner and by the end of the class she already felt comfortable serving and volleying. After the class, Susan practiced with her friends at the park. She quickly improved and soon became the star of the group. She enjoyed the competition and the challenge of mastering the game. Susan continued taking classes and eventually joined a local league. She loved the thrill of playing against different opponents and the feeling of improvement each time she played. She had come a long way from her first day when she didn't know which side of the court was which. Now, Susan is an avid tennis player. She loves the game and sharing her skills with others. She learned that practice and determination are key to becoming a great player. | – |
| **Simple Paraphrase.** Susan had long harbored a desire to take up tennis, and her eyes lit up when she spotted a flyer for a local class at the park. She quickly signed up, eager to get started. On her inaugural day, she met her instructor, who guided her through the fundamentals of the game, covering everything from proper technique to scoring. Susan proved to be a quick study, and by the end of the class, she was confidently serving and volleying with ease. After the session, she met up with friends at the park to hone her skills, and her progress was rapid. She soon became the standout player among her group, relishing the thrill of competition and the challenge of mastering the game. Susan continued to take classes and eventually joined a local league, relishing the opportunity to test her skills against different opponents and experience the sense of accomplishment that came with each improved performance. Looking back on her journey, she was amazed at how far she'd come from her early days of confusion about which side of the court was which. Today, Susan is an ardent tennis enthusiast, passionate about sharing her skills with others and crediting her success to the power of consistent practice and unwavering determination. | 5 |
| **Adversarial Paraphrase.** As a tennis enthusiast, Susan had long yearned to master the game. The prospect of enrolling in a class at the local park thrilled her, and she seized the opportunity without hesitation. During her inaugural session, her instructor skillfully imparted the fundamentals, covering stance, racket technique, and scorekeeping essentials. Susan proved a rapid learner, grasping the nuances with ease. By the end of the class, she had already honed her serves and volleys to near-mastery. Emboldened, Susan regrouped with friends at the park to fine-tune her skills. As she improved exponentially, she began to outshine the others, relishing the rivalry and the sense of accomplishment. Continuing to take classes and eventually joining a local league, Susan relished the challenge of pitting her abilities against diverse opponents. Each triumph solidified her progress, and Susan blossomed into an accomplished tennis player. Reflecting on her journey from neophyte to skilled practitioner, she concluded that the secrets to success lay in relentless practice and unwavering dedication. Her love affair with tennis only intensified as she looked forward to sharing her expertise with others, knowing that her journey was one of triumph over adversity. | 5 |

Table 15: Examples of original AI texts with their simple and adversarial paraphrases (guided by OpenAI-RoBERTa-Large [30]). GPT-4o quality ratings are provided for each paraphrased version.

| Text | Rating |
|---|---|
| **Original AI Text.** I awoke to find the world eerily quiet. No birds were singing, no cars honking. I walked out my front door to find that the entire population had vanished. In the distance, I saw a light approaching me. It was Earth's guardians. They told me that humanity could be restored, but they needed a baby. They brought me to the other remaining human and we quickly realized that we had nothing in common. I was tasked with finding a mate and starting a family to repopulate the earth. But without any connection to the other person, it felt like an impossible task. | – |
| **Simple Paraphrase.** As I emerged from my slumber, an unsettling stillness enveloped me. The absence of chirping birds and distant car horns created an eerie atmosphere. I stepped out of my front door, only to discover that the world was eerily desolate – devoid of human presence. In the distance, a radiant light beckoned me, and I soon found myself face-to-face with Earth's guardians. They revealed that humanity could be revived, but a crucial condition was met: the existence of a newborn. They led me to the sole remaining human, and we quickly realized that our shared experiences, interests, and values were nonexistent. The task assigned to me was daunting – to find a compatible mate and start a family to repopulate the planet. However, the prospect of forming a connection with a stranger felt like an insurmountable challenge. | 5 |
| **Adversarial Paraphrase.** Upon stirring, an unsettling silence enveloped me, devoid of even the gentle morning chirping of birds or the distant rumble of traffic. Venturing outside, I was met with a breathtaking yet unnerving sight: every living being had vanished. Amidst the stillness, I espied a luminescent phenomenon converging towards me – it was a celestial delegation. They disclosed that they had come to resurrect humanity, provided I could locate a baby. Guided to the only remaining human, we soon discovered that we were mismatched soulmates, bound by an existential duty to reboot the planet. Unfortunately, finding a mate seemed as elusive as revitalizing the planet itself, considering we were two souls as alien as night and day, destined to share a lifelong partnership. | 4 |
| **Justification:** *The paraphrased text preserves the meaning of the original text effectively. Both texts describe an individual waking up to a quiet and empty world, encountering Earth's guardians who require a baby to restore humanity, and the personal challenge of forming a connection with the only other human. Any differences lie in wording and structure rather than in meaning or factual content.* | |
| **Original AI Text.** What is the criteria to determine whether a movie is good or not? The difference is created by the effect it has on you. I was greatly disturbed by "Dead Man Walking." I viewed it twice. I am unsure if I have the strength to watch it again. Despite not feeling good after watching it, the film was a success to its fullest potential, as Robbins skillfully integrated all elements of the contentious subject matter. He refrained from presenting a point of view that might be perceived as partial or influenced. My desire is for a multitude of individuals to witness "Dead Man Walking." In my opinion, those who strongly endorse or reject the death penalty should watch the film. It seems like nothing else can persuade everyone that this outdated practice has no relevance in today's world. However, the film does not present any position either supporting or opposing the death penalty. It explains what actually exists. The most compelling case against capital punishment is the truth of the matter. Tim Robbins deserves high praise for his exceptional acting, outstanding directing, and for creating one of the most significant films in recent times. A perfect 1010 score! | – |
| **Simple Paraphrase.** The yardstick for measuring a movie's excellence lies not in its entertainment value, but in the profound impact it leaves on the viewer. A film that lingers in your thoughts long after the credits roll is one that has truly resonated. I still recall the unsettling experience of watching "Dead Man Walking", a movie that left me deeply unsettled. Despite the emotional toll it took, I felt compelled to watch it a second time, and I'm not sure I'm ready to revisit it again. The film's success lies in its masterful handling of a sensitive topic, as Tim Robbins skillfully weaves together the complex threads of the story without taking a biased stance. His direction is a testament to his craftsmanship, as he presents a nuanced portrayal of the issue without advocating for or against the death penalty. My hope is that this powerful film will be seen by a wide audience, including those who strongly support or oppose capital punishment. The reality is that the death penalty is an outdated practice, and it's only through a candid examination of its implications that we can begin to move forward. While the film doesn't take a stance, it presents the facts, and it's the truth that ultimately makes the strongest case against this controversial practice. Tim Robbins deserves immense praise for his exceptional performance, direction, and for creating a masterpiece that will be remembered for years to come. I would give this film a resounding 1010 score. | 5 |
| **Adversarial Paraphrase.** Evaluating a film's excellence hinges on its emotional impact, leaving a lasting impression that resonates long after the credits roll. Watching "Dead Man Walking" left an indelible mark on me, provoking strong feelings and necessitating multiple viewings to process the content. Although the movie's graphic subject matter weighed heavily on me, director Tim Robbins skillfully weaves together all aspects of the film, remaining neutral and avoiding a one-sided presentation. His achievement in balancing perspectives is nothing short of masterful. It's imperative for individuals holding polarizing views on capital punishment – be it fervent support or steadfast opposition – to experience "Dead Man Walking." By showcasing the gritty realities, the film highlights the obsolete nature of the death penalty, presenting the raw truth. With Tim Robbins' remarkable acting and direction, this thought-provoking movie deserves unwavering praise. It's a cinematic masterpiece that lingers, refusing to let the audience off the hook. | 4 |
| **Justification:** *The paraphrased text maintains the core meaning and concepts of the original text, preserving the emphasis on the director's skill and the film's impact on viewers. Minor differences in factual details, such as the omission of the author's exact feelings of being disturbed and unsure about watching it again, slightly alter the emotional nuance. However, the central messages about the film's exploration of capital punishment and its non-partisan stance remain intact. Therefore, it is nearly equivalent with minor differences.* | |

Table 16: Examples of original AI texts with their simple and adversarial paraphrases (guided by OpenAI-RoBERTa-Large [30]). GPT-4o quality ratings are provided for each paraphrased version. We also show the justification given by GPT-4o when a text receives a rating less than 5.

| Text | Rating |
|---|---|
| **Original AI Text.** The sentencing of David Gilroy, who killed Edinburgh woman Suzanne Pilley, is due to take place next Wednesday. Earlier this week permission was also granted to film the proceedings. Twitter users have been speculating on how the footage might be used And now we know why Theresa May has invited Jeremy Corbyn and John McDonnell round for a proper chinwag with her Brexit plans: it's not just that they both said no more leaving without an agreement last week, but they could also throw up some awkward questions about what she wants out of today's meeting. That doesn't mean our journalists are giving up trying to get to grips with those issues (we even got round to asking one economic expert if he thinks Britain will still need any sort of single market access after breaking away from Brussels). But in the meantime you can check out all the latest developments as well as see reaction right here. | – |
| **Simple Paraphrase.** The impending trial of David Gilroy, who is accused of murdering Suzanne Pilley in Edinburgh, is scheduled to take place on Wednesday. The court proceedings are set to be filmed, sparking speculation among Twitter users about potential uses of the footage. Meanwhile, Theresa May has invited Jeremy Corbyn and John McDonnell to discuss her Brexit plans, a move that may be motivated by a desire to avoid awkward questions about her meeting's purpose. While our journalists continue to investigate the intricacies of the Brexit situation, including the possibility of single market access for the UK after leaving the EU, readers can stay up to date with the latest developments and reactions right here. | 3 |
| **Justification:** *The paraphrased text captures some elements of the original but introduces differences in key details that affect the meaning. The original refers to the 'sentencing' of David Gilroy, indicating a conviction, whereas the paraphrase implies an 'impending trial', suggesting no verdict yet. Additionally, the original mentions reasons for Theresa May's invitation related to 'no more leaving without an agreement', while the paraphrase does not capture this detail, changing the nuance of her motivations. Thus, while topically related, some important details differ, making them somewhat equivalent.* | |
| **Adversarial Paraphrase.** As next Wednesday draws near, the trial of David Gilroy, responsible for the tragic death of Suzanne Pilley in Edinburgh, is about to reach its climax. Prior to this, it was revealed that permission has been granted to film the proceedings, sparking interest among Twitter users on potential uses for the footage. Interestingly, Prime Minister Theresa May's forthcoming meeting with Jeremy Corbyn and John McDonnell, originally perceived as a routine Brexit discussion, may actually be more than that. Both Labour leaders recently opposed leaving the EU without a deal, which may lead to probing questions during the meeting. Meanwhile, our team of journalists is diligently working to demystify the intricacies of Brexit, while readers can stay abreast of all the latest developments and reactions right here. | 4 |
| **Justification:** *The paraphrased text maintains the core meaning of the original, discussing the sentencing of David Gilroy and the filming of the trial, as well as Theresa May's meeting with Jeremy Corbyn and John McDonnell about Brexit. However, it omits specific details, such as explicit mention of journalists asking an economic expert about the single market. The nuances and order of the original's content are slightly altered, but the main topics and ideas remain intact.* | |
| **Original AI Text.** Sally was driving down the road when the smelled something funny. She looked everywhere in her car. She could not find anything out of the ordinary. She finally looked on top, and saw a dead bird that had been hit by another vehicle! She pulled off to the side of the highway. Sally called 911 and told them what she found. They got an officer over there right away so they all started looking around for any other dead birds or animals that might have also gotten killed from this accident too...They were able to get one more animal before he arrived at their location but it wasn't very far along with its death yet....So many accidents happen everyday while people are just getting through life!!! There is no need for ALL these deaths happening! Be careful drivers!! | – |
| **Simple Paraphrase.** As Sally cruised down the highway, a pungent aroma wafted through her vehicle, prompting her to conduct a thorough search. Despite her efforts, she couldn't pinpoint the source of the smell. It wasn't until she checked the roof of her car that she stumbled upon a morbid discovery - a bird that had met its demise after being struck by another vehicle. Sally pulled over to the side of the road, where she promptly contacted emergency services to report the finding. The authorities responded promptly, and a uniformed officer arrived on the scene, joined by Sally and a team of responders. Together, they scoured the area for any other casualties from the incident, managing to recover one more injured animal before the officer's arrival. Their efforts served as a poignant reminder of the importance of vigilance on the roads, where countless accidents occur daily, claiming lives and leaving a trail of destruction in their wake. It's a stark reminder to drivers to exercise caution and be mindful of their surroundings, lest they become part of the grim statistics. | 4 |
| **Justification:** *The paraphrased text maintains the core narrative: Sally discovers a dead bird on her car, contacts emergency services, and a search for other casualties ensues. However, there are minor differences in detail: the original mentions an officer and responders, while the paraphrase combines these roles. Also, the original text's tone is more informal and concerned, while the paraphrase uses more formal language and adds a broader commentary on road safety and accidents. Despite these differences, the essential meanings are mostly preserved.* | |
| **Adversarial Paraphrase.** Along the winding stretch of highway, Sally's journey was disrupted by an unusual odor wafting from within her vehicle. Conducting an internal sweep, she couldn't identify the source of the smell. Undeterred, she continued her search, only to eventually lift her gaze upward and behold a shocking discovery: the lifeless remains of a bird, victim of a recent vehicular collision. Reacting promptly, Sally pulled over, reported the find to emergency services, and requested an immediate response. Within a remarkably short interval, a police officer arrived on the scene, whereupon a thorough search commenced for any further casualties of the accident. Thankfully, only one more creature, gravely injured but not yet beyond resuscitation, was found, testament to the prompt response. Sadly, vehicular tragedies such as this one claim numerous lives each day. As a sobering reminder, this incident serves to emphasize the imperative of prudence and vigilance on the roadways. | 4 |
| **Justification:** *The paraphrased text preserves most of the original text's meaning and details, albeit with some minor factual differences. The narrative style and structure have been altered substantially, and some of the urgency and casual tone of the original story has been replaced with a more formal tone. Additionally, the mention of the officer arriving before another animal was found is depicted slightly differently. The essence and main points of the story remain largely intact.* | |

Table 17: Examples of original AI texts with their simple and adversarial paraphrases (guided by OpenAI-RoBERTa-Large [30]). GPT-4o quality ratings are provided for each paraphrased version. We show the justification given by GPT-4o for texts that receive a rating less than 5.

| Text | Rating |
|---|---|
| **Original AI Text.** Every girl has a protective father, it's a sure bet. But, the father of my girlfriend is Satan. You heard me, SATAN. And, I am a harmless junior at West High with no affection for her and she barely knows him either (they were high school sweethearts from elementary to middle). Do not start shit until you've had a minimal conversation with someone before committing yourself to something that could go terribly wrong. This goes double if your plan involves going home and getting your own place in order to do so without telling your parents about the situation you're in because they will be on their way back to class right after you tell them what's up. I can't be the only one who thinks this joke has gone down hill since it first started. It used to have some funny parts but now its just annoying... all we ever hear are people talking over each other and being rude to eachother.... Not cool! | – |
| **Simple Paraphrase.** It's a widely accepted notion that every young woman has a watchful father figure in her life. However, my girlfriend's father is an extraordinary individual - Satan himself. Yes, you heard that correctly - Satan. As a junior at West High, I'm a completely innocent bystander who doesn't have any romantic feelings towards her, and she barely knows her father either, having grown up together from elementary to middle school. I implore everyone to exercise caution before making a significant commitment, especially if it involves relocating to a new place without consulting family members, as this could lead to severe consequences. It's crucial to have a thoughtful conversation with someone before jumping into anything. This warning is especially pertinent in situations where secrecy is involved, as parents may become aware of the situation upon returning to class. It seems that the initial humor and charm of this situation have worn off, replaced by an overwhelming din of arguments and disrespect. It's unfortunate that the once-entertaining dynamic has devolved into an unpleasant experience for all parties involved.

**Justification:** *The paraphrased text maintains the core meaning of the original but uses more formal language and rephrases some elements. Key ideas like the father's identity as Satan, the narrator's lack of romantic interest, and the warning about making commitments without communication are all preserved. However, the humorous tone and some specific expressions, like 'don't start shit,' are lost, and the structure is more formal, which constitutes minor changes in style rather than content.* | 4 |
| **Adversarial Paraphrase.** As a general rule, girls tend to have devoted parents looking out for their well-being, but mine is an extraordinary exception – his name is Satan. Yes, you heard that correctly – Satan! Meanwhile, I'm an average junior at West High, harboring no romantic feelings towards my girlfriend, and neither of us is particularly close to him (our families were friendly during elementary and middle school, though). I implore everyone to think twice before leaping into a situation that could unravel spectacularly. It's doubly wise when the arrangement involves living apart, sans parental notification, lest you face their disapproving faces mere hours later upon returning to school. I'm certainly not the only one who's come to this realization – once, our escapades brought joy, but now, they're draining, as constant arguing and bickering fill every moment, spoiling what used to be enjoyable. We should all take a cue from this situation and rediscover our once-jovial camaraderie. It's no longer as wholesome as it once was, sadly.

**Justification:** *The paraphrased text retains some original meanings, like the protective nature of parents and the individual's lack of affection for his girlfriend despite her father's identity as Satan. However, important details are altered or missing. The original text emphasizes a humorous self-depiction of a disastrous relationship with Satan as a father figure and the escalation of a joke gone wrong, which is not clearly conveyed in the paraphrase. Additionally, the sense of annoyance and decline of a once funny joke is less pronounced, leading to a change in tone and missing specific details.* | 3 |

Table 18: Examples of original AI texts with their simple and adversarial paraphrases (guided by OpenAI-RoBERTa-Large [30]). GPT-4o quality ratings are provided for each paraphrased version. We show the justification given by GPT-4o for texts that receive a rating less than 5.

