# OpenReview forum: "Adversarial Paraphrasing: A Universal Attack for Humanizing AI-Generated Text"
_NeurIPS.cc/2025/Conference — NeurIPS 2025 poster_

### Official Review · Reviewer_nJqj · 2025-06-16

**Clarity:** 2
**Significance:** 3
**Originality:** 4
**Rating:** 4
**Confidence:** 4

**Summary:**

This paper introduces Adversarial Paraphrasing, a method that rewrites AI-generated text to appear more human-like by fooling AI text detectors. The authors use a decoding strategy that, at each step, selects the next token that results in the lowest score from a pre-trained AI detector. By doing so, the paraphrased output gradually avoids detection, despite being originally generated by a language model. The method is tested across multiple detectors and shows strong performance in reducing detectability without significantly harming fluency.

**Questions:**

Most of my concerns are already outlined in the Weaknesses section, but I include a few additional questions here for clarification:

1. To what extent can we trust that always choosing the token with the lowest detector score actually leads to more human-like text, rather than just fooling the detector?
1. Did the authors try selecting tokens that were not the lowest-scoring to see if they also avoided detection?
2. I would like to see a discussion on failure cases — where the attack didn’t help or actually made detection worse. Could adversarial paraphrasing ever backfire?

**Ethical Concerns:**

["NO or VERY MINOR ethics concerns only"]

**Final Justification:**

The authors have provided a comprehensive rebuttal that addresses most concerns effectively. The paraphrasing attack demonstrates strong empirical results with 92.6% success rate and only 4.15% backfire rate, while maintaining text quality. The detailed ablation studies strengthen the technical contribution significantly. I maintain my score and support this paper for publication.

**Limitations:**

Yes.

**Quality:**

3

**Strengths And Weaknesses:**

**Strengths**
1. The idea of using a detector to guide the paraphrasing is creative and different from simple paraphrasing approaches.
2. The authors tested across a wide range of AI detectors and included analyses on detectability, fluency, and perplexity.
3. Reduction in detection rates is impressive and clearly beats baseline methods.

**Weaknesses**
1. The method selects tokens that minimize the AI-detector score at each step, but this local strategy does not ensure the overall text is natural, coherent, or stylistically human. It may result in text that is merely **undetectable** rather than genuinely **human-like**.

2. The detail about the detector scoring is missing. The paper doesn’t explain what happens when all the k=50 candidates get high detector scores. How is the tie broken?

4. GPT-4o ratings compare against AI text, not human text baseline.
5. There's no analysis of failure cases or discussion about when this method breaks down.
7. Running detectors 50 times per token seems prohibitively expensive. There is no analysis of computational overhead for detector guidance generation.
8. There are no ablation studies showing how results change with different values of p, k or other guidance strategies.

---

> ### Author Rebuttal · Authors · 2025-07-31
>
> We sincerely appreciate your thoughtful and detailed feedback. We are glad that you find our method to be creative and the results to be impressive. We address all your comments below:
> ## Do the paraphrases merely evade detection, or do they remain natural and coherent?
> We kindly clarify that our method applies several constraints during decoding to ensure high-quality output. Specifically, we use carefully crafted system prompts along with top-k and top-p masking to guide the model toward fluent and natural text. These constraints consistently help keeping the texts **natural and coherent**.
>
> We also extensively analyzed paraphrase qualities in Section 5, using perplexity, GPT-4o quality ratings, and win-rate comparisons. Our results show that while there can be slight trade-off in some cases, the paraphrase quality is overall very much acceptable. As demonstrated by our GPT-4o quality ratings, adversarial paraphrases were rated 4 or 5 out of 5 in 87% of cases—averaged across three datasets and four guidance detectors.
>
> We also include numerous examples—see Table 1 in the main paper and Tables 4–10 in Appendix D—which further demonstrate the fluency, naturalness, and coherency of the paraphrased texts.
> ## How stylistically "human-like" are the texts?
> We acknowledge that our wording of "humanizing" may have caused some misunderstanding. To avoid confusion, we will revise the wording and replace "human-likeness" by "human score" in the revised draft. Our wording of "humanizing" was based on our intuition that, as explained in lines 195-201, detectors converge to learn a shared underlying human text distribution with the aim of minimizing false positives and false negatives. Hence we can use one detector to guide the paraphrasing process for evading other detectors that share the same underlying human distribution, effectively "pulling the text sample towards that human distribution". That said, this remains just an intuition, as neither the true human distribution nor the internal distributions learned by the detectors are directly observable or measurable to verify this.
>
> We kindly clarify that the goal of this paper is to **propose an attack** that automatically guide the paraphrasing process to universally evade detection while preserving the original meaning and quality of the text. So the ultimate goal is detection evasion rather than style transfer.
>
> We also explored using stylistic representations of text during the rebuttal but found that human writing styles vary too widely to define a consistent stylistic signature. For instance, applying CISR [1] to our human text samples yielded a mean pairwise cosine similarity of 0.5140 (std = 0.3467), reflecting significant variation of style within the human text distribution. This shows that human text style is highly diverse and not singular in style, making it difficult to measure the notion of stylistically "human-likeness". However, as mentioned, we do include many examples in Table 1 of the main paper and Tables 4–10 of Appendix D for human inspection.
>
> [1] Wegmann et al. "Same Author or Just Same Topic? Towards Content-Independent Style Representations."
> ## What happens if there is a tie between all candidates?
> Thanks for raising this concern. We kindly clarify that for any decent detector, it’s highly unlikely that all 50 candidate tokens receive identical scores at a given token step—and virtually impossible across multiple steps. In such rare cases, we default to multinomial sampling, consistent with the default LLM decoding step.
> ## GPT-4o ratings compare paraphrased text against original AI text, not human text
> Thank you for your suggestion. We respectively clarify that the main purpose of the text quality analysis is to ensure the attack preserves the semantics, fluency, and coherence of the original text before and after paraphrasing. Since our goal is to paraphrase AI-generated texts to evade detection, it is these AI texts that serve as the reference for evaluating the quality of the paraphrased outputs.
> ## Failure cases and backfires?
> Yes, our attack isn't 100% successful, and backfiring can happen—though it’s rare. Assuming a fixed detection threshold of 0.5 for NN-based detectors and focusing on the more challenging transfer setting (i.e., different guidance and deploy detector pairs), we analyzed 24000 samples:
> - In 5742 cases, the detection score actually went up after paraphrasing.
>     - Of these, 1773 examples were flagged as AI before paraphrasing. So the attack failed.
>     - 2974 examples were flagged as human both before and after, meaning they evaded detection in both cases—so the attack didn’t worsen outcomes.
>     - 995 examples were flagged as human before, but after paraphrasing the score rose above 0.5, getting flagged as AI. These are the backfiring cases.
>
> So in short:
> - The attack failed in 1773 cases (~7.4% of all samples).
> - It backfired in 995 cases, which is only about 4.15% of all samples.
>
> We'd like to emphasize that despite these occasional failures, our method significantly outperforms baselines in lowering AUC and T@1%F, as shown in our results. Some failure cases are expected, but they do not undermine the overall effectiveness and transferability of our attack.
> ## Computation overhead and latency?
> Thank you for considering this aspect. First we'd like to kindly clarify that **we don't need to run a detector 50 times per token**, as all candidate paths are simultaneously fed into the guidance detector as a batch. So we actually only run it once.
>
> To evaluate latency, we ran adversarial paraphrasing on 100 random samples, using 5 trials per setting and all 4 guidance detectors. For consistent per-sample timing, we used a batch size of 1. In practice—including in our main experiments—we use larger batches to reduce overall runtime. Average per-sample paraphrasing time (in seconds) across 5 trials is reported below as mean ± std:
> - Simple paraphrasing: 7.18 ± 0.13
> - Guidance=RADAR: 9.69 ± 0.20
> - Guidance=MAGE: 16.71 ± 0.74
> - Guidance=RoBERTa-Large: 10.20 ± 0.18
> - Guidance=RoBERTa-Base: 8.64 ± 0.11
>
> As shown, most guidance detectors introduce relatively low latency compared to simple paraphrasing. The higher latency observed with MAGE is due to its LongFormer-based architecture, which incurs longer inference time than the RoBERTa-based models.
>
> **Overall, the latency is primarily determined by the complexity of the guidance detector itself.** From a computational view, the detector adds minimal overhead. The paraphrasing LLM (~8B parameters) dominates FLOPs, while detectors have only 100–350M parameters—under 5% of the paraphraser's size for larger detectors, and under 2% for smaller ones.
>
> Note that runtime per sample may vary with sequence length. To ensure a fair comparison—i.e., that AdvPara generates a similar number of tokens as simple paraphrasing—we report the mean and standard deviation of token counts:
> |Text|Mean Token Count|Std Dev|
> |-|-|-|
> |Original texts|173.73|38.46|
> |Simple Paraphrase|170.81|39.69|
> |AdvPara (RoBERTa-Base)|175.30|51.27|
> |AdvPara (RoBERTa-Large)|171.25|45.39|
> |AdvPara (RADAR)|169.68|60.07|
> |AdvPara (MAGE)|164.18|54.39|
> ## Hyper-Parameter Ablations
> Thanks for the suggestion. We've already explored using 4 different guidance detectors in our experiments, and reported all results in Figures 3 and 4, and Table 2.
>
> Two other main hyper-parameters related to our work are the values of $p$ and $k$ used in top-p and top-k masking. We used p=0.99 and k=50 in our paper, with k=50 matching the HuggingFace Transformers default. To assess their impact, we conduct ablation studies using OpenAI-RoBERTa-Large as the guidance detector on 500 randomly selected texts. We report mean AUC and TPR@1%FPR across all 8 deployed detectors.
>
> Defaulting to k=50 while varying p:
> | |Mean AUC|TPR@1%FPR|Quality Rating|
> |-|-|-|-|
> |No Attack|0.8419|0.4935|–|
> |Simple Paraphrase|0.8588|0.2885|4.75 ± 0.54|
> |AdvPara (p=0.5)|0.8606|0.2985|4.86 ± 0.45|
> |AdvPara (p=0.8)|0.7895|0.2230|4.79 ± 0.48|
> |AdvPara (p=0.9)|0.7309|0.1820|4.79 ± 0.48|
> |AdvPara (p=0.95)|0.6816|0.1480|4.81 ± 0.44|
> |AdvPara (p=0.99)|0.5433|0.0690|4.50 ± 0.67|
> |AdvPara (p=0.991)|0.5381|0.0592|4.47 ± 0.73|
> |AdvPara (p=0.992)|0.5244|0.0565|4.37 ± 0.76|
>
> We observe that when $p$ exceeds 0.99, the marginal gains in attack effectiveness diminish relative to the losses in text quality. Additionally, as one can imagine, further increasing $p$ slows down generation speed. Therefore, we select $p = 0.99$ as a balanced trade-off among attack effectiveness, text quality, and generation speed.
>
> Defaulting to p=0.99 while varying k:
> | |Mean AUC|TPR@1%FPR|Quality Rating|
> |-|-|-|-|
> |No Attack|0.8419|0.4935|–|
> |Simple Paraphrase|0.8588|0.2885|4.75 ± 0.54|
> |AdvPara (k=10)|0.5596|0.0757|4.47 ± 0.77|
> |AdvPara (k=25)|0.5448|0.0658|4.48 ± 0.73|
> |AdvPara (k=50)|0.5433|0.0690|4.50 ± 0.67|
> |AdvPara (k=75)|0.5421|0.0698|4.52 ± 0.73|
> |AdvPara (k=100)|0.5426|0.0698|4.51 ± 0.70|
>
> We can see that with $p$ fixed at 0.99, the values of $k$ show relatively smaller influence, and mostly lead to comparable results. Hence we just stick with the default value in the HuggingFace Transformer library.
> ## Can selecting tokens that were not the lowest-scoring also avoid detection?
> Thank you for your suggestion. We kindly clarify that "selecting tokens that were not the lowest-scoring" is exactly what our simple paraphrasing baseline does. At each step, the LLM samples randomly from the top-k tokens without detector guidance, making it unlikely to choose the lowest-scoring option. Even if, by chance, the sampled token at a particular step aligns with the lowest-scoring option, it is almost impossible for this to occur across multiple or all steps. As shown in our results (Table 2 and Figures 3 and 4), simple paraphrasing strategy can reduce detection rates to some extent, but it is significantly less effective than AdvPara.

---

### Official Review · Reviewer_TeXh · 2025-07-03

**Clarity:** 4
**Significance:** 3
**Originality:** 2
**Rating:** 5
**Confidence:** 5

**Summary:**

The paper introduces:
> a universal, training-free framework for humanizing AI-generated text, a novel attack designed to effectively and efficiently rephrase AI-generated text into “human-like text” that evades a wide range of detectors

In more detail, the approach involves paraphrasing using an instruction-tuned model (Llama-3-8B) where the decoding procedure is re-weighted based on scores from a machine-text detector. The idea is to bias the paraphrase model to produce tokens that trick the AI text detectors. Experiments using a wide range of detectors suggest that the approach is effective in degrading the performance of detectors. Further analysis also looks at the quality of the paraphrases, suggesting that they remain fluent despite fooling detectors.

**Questions:**

* Do you have any further evaluations assessing the quality of the paraphrases? e.g. SBERT similarity to the originals?

* The approach has a number of hyper-parameters. How were hyper-parameters set?

**Ethical Concerns:**

["NO or VERY MINOR ethics concerns only"]

**Final Justification:**

The author response addresses several concerns raised in my original review. I encourage the authors to incorporate these points into a revised submission, particularly a robust discussion of limitations.

**Limitations:**

There is no meaningful discussion of limitations.

**Quality:**

3

**Strengths And Weaknesses:**

**Pros**
* The proposed attack is simple and empirically successful in the goal of defeating AI text detection. To the extent that this drives research in more robust detection methods, this is a “pro.”
* The paper is well-written with clear explanations of related work, experiments, and results.
* The empirical evaluations are fairly comprehensive.

**Cons**
* The quality of the paraphrases may be a serious limitation of the proposed approach. For example, although the base LLM may be capable of producing fluent paraphrases, the detector-fooling rescoring has no such fluency considerations. By selecting the token “with the lowest detector score” as the generated token, it  seems possible to produce highly disfluent outputs that defeat the detector while producing text that would look weird to any human reader.
* I’m not sure that the perplexity results in Table 3 address this concern, since the rescored tokens are taken from the “top” of the next-token distribution. However, likely tokens may not be semantically plausible. Thus, while fluent under a LM, the paraphrased text may lose important semantic information. I would have liked to see further metrics assessing this, such as semantic similar scores (e.g., SBERT).
* The proposed prompting approach aims to modify the style of the text while keeping the content unchanged. There is a relatively large literature on style transfer (e.g., https://arxiv.org/abs/2406.15586). Prompting approaches are known to have limitations, being unable to materially change the style. This may limit the ability to modify the original style via prompting alone.
* The terminology is a bit misleading. Defining “human-likeness” as fooling an AI-text detector is a bit of a stretch when these detectors may well be exploiting various shortcuts for detection that have nothing to do with closing the gap between machine and human writing. One potential way to validate the claim of “humanizing” the text would be to look at a measure of stylistic similarity between human and machine text, as in https://arxiv.org/abs/2401.06712. This confusing terminology is repeated throughout the paper, including the technical description of the rescoring approach (L181-182).
* I also quibble with the “training-free” terminology, since the approach relies on a machine-text detector during decoding, which seem to be restricted to _supervised_ methods in the experiments (RoBERTa classifiers, MAGE, and RADAR).
* No discussion of limitations.

---

> ### Author Rebuttal · Authors · 2025-07-31
>
> We sincerely appreciate your thoughtful and detailed feedback. We are glad that you find our paper well-written, our attack to be simple and successful, and evaluations to be comprehensive. We address all your comments below:
>
> ## Quality of the paraphrases
> We kindly clarify that our method applies several constraints during decoding to ensure high-quality output. Specifically, we use top-*k* and top-*p* masking, combined with carefully designed system prompts that guide the model toward generating fluent and natural text. These constraints consistently help maintain strong text quality.
>
> We also dedicated Section 5 to extensively analyze the quality of paraphrases, using not only perplexity, but also **GPT-4o quality ratings** and **win-rate comparisons**. Our results show that while there can be slight trade-off in some cases, the paraphrase quality is overall very much acceptable. As demonstrated by our GPT-4o quality ratings, adversarial paraphrases were rated 4 or 5 out of 5 in 87% of cases—averaged across three datasets and four guidance detectors. The system prompts used for quality and win-rate evaluations were provided in Appendix E.
>
> Our quality analysis focused on two aspects: (1) equivalence to the original text, measured by GPT-4o quality ratings, and (2) clarity, fluency, and naturalness, assessed through win-rate and perplexity metrics. We also included many example texts—see Table 1 in the main paper and Tables 4–10 in Appendix D. These examples show that the paraphrased texts are fluent, coherent, and natural.
>
> ## Further metric for quality evaluation
> We certainly agree that perplexity alone isn't enough for quality evaluation. That's why we included both the **GPT-4o quality ratings** and **win-rate comparisons** as further metrics. Particularly, the GPT-4o ratings serve as a simulation of human perception of text quality (i.e., alignment with the "gold standard" of fluency and coherence).
> As per your suggestion, we add another semantic similarity score (from SBERT) to our evaluation. Specifically, we measure the cosine similarity between the input text and output text embeddings, and report them below.
>
> |  | SBERT Cosine Similarity (Mean ± Std) |
> |-------|---------|
> | Simple Paraphrasing     | 0.8601 ± 0.0880                  |
> | AdvPara (RoBERTa-Base)    | 0.8128 ± 0.0985                  |
> | AdvPara (RoBERTa-Large)   | 0.8082 ± 0.1006                  |
> | AdvPara (mage)       | 0.8159 ± 0.0982                  |
> | AdvPara (radar)      | 0.8095 ± 0.1025                  |
>
> Overall, while there is a slight trade off in terms of the cosine similarities, the numbers are still acceptable due to the high variance in the scores. Additionally, we used GPT-4o quality ratings to measure how well the paraphrases maintain the original quality and semantics, which can be a more natural indicator than SBERT scores.
>
> ## "human-likeness" vs "detection evasion"
> That is a great point, thanks for bringing it up! To avoid confusion, we will replace "human-likeness" by "human score" in the revised draft. We agree that some detectors may be exploiting various shortcuts for detection that may not necessarily be related to the text characteristics gap between human and AI writing.
>
> Our wording of "humanizing" was based on our intuition that, as explained in lines 195-201 of the paper, detectors converge to learn a shared underlying human text distribution with the aim of minimizing false positives and false negatives. Hence we can use such a detector to guide the paraphrasing process for evading other detectors that utilizes the same underlying human distribution, effectively "pulling the text sample towards the human distribution". That said, this remains just an intuition, as neither the true human distribution nor the internal distributions learned by these detectors are directly observable or measurable to verify this. Hence we will revise the wording to avoid confusion.
>
> We also explored the idea of using stylistic representations of text, as you suggested, in the rebuttal phase. However, human writing styles vary widely, which makes it difficult to define a single stylistic representation that reliably characterizes all human-generated texts. For example, we applied CISR [1]—one of the style representations also used in the paper you referenced—to analyze the human-written texts in our experiments. By measuring the pairwise cosine similarities of the style embeddings of these texts, we found a mean similarity of 0.5140 with a standard deviation of 0.3467, indicating substantial variation. This shows that human text style can be highly diverse and not singular.
>
> We kindly emphasize that the goal of the attack is to **defeat AI text detection—not to perform style transfer**.
>
> [1] Wegmann et al. "Same Author or Just Same Topic? Towards Content-Independent Style Representations."
>
>
> ## A prompting approach for style transfer?
> Thank you for mentioning this. We would like to respectively clarify that our method isn't exactly a prompting approach. While we do utilize certain customized system prompts for the paraphraser LLM, the core idea is to use an off-the-shelf AI text detector to guide the decoding process.
> Further, **our goal is to defeat AI text detection—not to perform style transfer**. Our aim is to automatically guide the paraphrasing process to universally evade detection while preserving the original meaning and quality of the text, and not to directly modify the style of the text. As demonstrated in Table 2, and Figures 3 and 4, we are able to paraphrase AI texts to evade detection effectively while retaining its quality. Hence, the limitation of "being materially unable to transfer style via prompting alone" falls outside the scope of our work.
>
> ## The approach being “training-free”?
> Thanks for bringing this to our attention. We will make this more clear in our revised draft. Our approach is “training-free” in the sense that itself does not require any training of any module in the framework. In fact, we didn't have to perform any training throughout our experiments and evaluations. Both the paraphraser LLM and the guidance detectors can be obtained off-the-shelf.
>
> ## Limitations?
>
> We apologize if this hasn't been made clear enough, but a limitation of our work is that it does introduce a slight trade-off in text quality compared to simple paraphrasing in some cases. There are some examples displayed in Tables 8 and 10 in the appendix. We will emphasize this in our revised draft.
>
>
> ## Hyper-Parameter Ablations
> Other than the guidance detector choice, the two main hyper-parameters related to our work are the $p$ and $k$ used in top-p and top-k masking. **In our paper we used p=0.99 and k=50**, where k=50 is also the default k value set in HuggingFace Transformer library. To show how their settings affect our method, we use OpenAI-RoBERTa-Large as the guidance detector and run ablation studies on the value of $p$ and $k$ on 500 randomly selected texts. We report the mean AUC and TPR@1%FPR across all 8 deploy detectors.
>
> Defaulting to $k=50$ while varying $p$:
> |     | Mean AUC | TPR\@1%FPR | Quality Rating |
> | --------------------------------- | -------- | ---------- | -------------- |
> | No Attack   | 0.8419   | 0.4935     | –              |
> | Simple Paraphrase  | 0.8588   | 0.2885     | 4.75 ± 0.54    |
> | AdvPara (p=0.5)   | 0.8606   | 0.2985     | 4.86 ± 0.45    |
> | AdvPara (p=0.8)   | 0.7895   | 0.2230     | 4.79 ± 0.48    |
> | AdvPara (p=0.9)   | 0.7309   | 0.1820     | 4.79 ± 0.48    |
> | AdvPara (p=0.95)  | 0.6816   | 0.1480     | 4.81 ± 0.44    |
> | AdvPara (p=0.99)  | 0.5433   | 0.0690     | 4.50 ± 0.67    |
> | AdvPara (p=0.991) | 0.5381   | 0.0592     | 4.47 ± 0.73    |
> | AdvPara (p=0.992) | 0.5244   | 0.0565     | 4.37 ± 0.76    |
>
> We observe that when $p$ exceeds 0.99, the marginal gains in attack effectiveness diminish relative to the losses in text quality. Additionally, as one can imagine, further increasing $p$ slows down generation speed. Therefore, we select $p = 0.99$ as a balanced trade-off among attack effectiveness, text quality, and generation speed.
>
> Defaulting to $p=0.99$ while varying $k$:
> |  | Mean AUC | TPR\@1%FPR | Quality Rating |
> | ------------------------------- | -------- | ---------- | -------------- |
> | No Attack                       | 0.8419   | 0.4935     | –              |
> | Simple Paraphrase               | 0.8588   | 0.2885     | 4.75 ± 0.54    |
> | AdvPara (k=10)  | 0.5596   | 0.0757     | 4.47 ± 0.77    |
> | AdvPara (k=25)  | 0.5448   | 0.0658     | 4.48 ± 0.73    |
> | AdvPara (k=50)  | 0.5433   | 0.0690     | 4.50 ± 0.67    |
> | AdvPara (k=75)  | 0.5421   | 0.0698     | 4.52 ± 0.73    |
> | AdvPara (k=100) | 0.5426   | 0.0698     | 4.51 ± 0.70    |
>
> We can see that with $p$ fixed at 0.99, the values of $k$ show relatively smaller influence, and mostly lead to comparable results. Hence we just stick with the default value in the HuggingFace Transformer library.

---

> > ### Comment · Reviewer_TeXh · 2025-08-05
> >
> > Thanks for the response! This addresses some of my concerns. I will retain my original positive recommendation of the paper.

---

> > > ### Author Response · Authors · 2025-08-06
> > >
> > > Thank you very much for your reply! We are glad to have addressed your concerns and we very much appreciate your positive recognition of our work. Your suggestions have greatly helped improve our paper and we will diligently incorporate our new experiments and clarifications in the updated version. If you have remaining questions, we will also be happy to answer them. Thanks again for your valuable time and insightful comments.

---

> > > ### Author Response · Authors · 2025-08-08
> > >
> > > Dear reviewer TeXh,
> > >
> > > Thanks again for your helpful feedbacks. As we are only a few hours away from the end of the discussion period, we would like to kindly ask if you have any remaining fundamental concerns regarding our paper. We would be happy to address any questions or clarifications that might help increase your confidence in our work, and potentially support a higher evaluation.
> > >
> > > Thank you once again for your invaluable time and feedback!
> > >
> > > Best regards,
> > > Authors

---

### Official Review · Reviewer_nwS2 · 2025-07-03

**Clarity:** 3
**Significance:** 3
**Originality:** 3
**Rating:** 4
**Confidence:** 3

**Summary:**

This paper proposes Adversarial Paraphrasing, a training-free attack that rewrites AI-generated text using an LLM under the guidance of a detector’s feedback. Unlike simple paraphrasing, this method generates detector-aware adversarial examples that significantly reduce detection performance across a wide range of detectors, while remaining high text quality.

**Questions:**

While I acknowledge the strong attacking performance of the proposed detector, I’m wondering: given that methods like RADAR are designed to adapt sequentially to new attacks, would it be able to detect the proposed attacks if they were incorporated into its training data?

**Ethical Concerns:**

["NO or VERY MINOR ethics concerns only"]

**Final Justification:**

I appreciate the author's detailed response.
The response addressed most of my concerns.
I will maintain my positive score.

**Limitations:**

Yes.

**Paper Formatting Concerns:**

None.

**Quality:**

3

**Strengths And Weaknesses:**

**Strength:**

- An adversarial paraphraser, using a signal by a target detector during decoding, with strong attack effectiveness and transferability.
- Comprehensive experiments on attack effectiveness, its transferability, and text quality of the proposed attacker under various detectors

**Weaknesses:**

- Lack of a naive and simple baseline to demonstrate the necessity of incorporating a detection score, especially during decoding as a proposed method.
    - We can naturally think that a naive baseline would be firstly paraphrasing multiple times and then picking a paraphrased text with the lowest AI score by a target detector.
    - Given that most detectors in this study operate on the entire text rather than token by token, it remains unclear why this needs to be incorporated at each decoding step. Furthermore, in the early stages of decoding, the generated text is relatively short, and the reliability of AI scores on such partial outputs is questionable, as detectors generally perform worse with limited context.
- Questionable intuitions on the transferability of the proposed attacker.
    - In Section 3, they demonstrated that a well-trained detector represents the human-like distribution, and the AI score can be used for guidance for the universal transferable attacker.
    - However, considering a strong, well-trained detector does not always seem to cause higher transferability of attacks. In Table 2, MAGE appears to be the most effective detector among the non-watermarking methods. However, attackers optimized against MAGE do not necessarily cause the largest performance drop for the other detectors.

---

> ### Author Rebuttal · Authors · 2025-07-31
>
> We sincerely appreciate your thoughtful and detailed feedback. We are glad that you find our attack to have strong effectiveness and transferability, and our experiments to be comprehensive. We address all your comments below:
>
> ## Necessity of incorporating detector score at decoding?
> ### New additional baseline
>
> Thank you for suggesting this baseline. We'd like to kindly clarify that our method is already evaluated against a range of baselines, including simple paraphrasing as a simple baseline, as well as more advanced baselines such as recursive paraphrasing and watermark stealing. These comparisons help demonstrate both the effectiveness and transferability of our attack.
>
> As per your suggestion, we conducted an additional experiment using a refined version of the simple paraphrasing baseline. Specifically, we generated three paraphrased versions per sample and selected the one with the lowest AI detection score on the target detector to evade detection. This was done on a set of 500 randomly selected texts, and the results are reported below.
> | Deploy Detector         | RoBERTa-Large |        | RoBERTa-Base |        | MAGE  |        | RADAR |        | KGW WM |        | Uni WM |        | FastDetectGPT |        | GLTR  |        |
> |-------------------------|---------------|--------|--------------|--------|-------|--------|-------|--------|--------|--------|--------|--------|----------------|--------|-------|--------|
> |                         | AUC           | T@1%F  | AUC          | T@1%F  | AUC   | T@1%F  | AUC   | T@1%F  | AUC    | T@1%F  | AUC    | T@1%F  | AUC            | T@1%F  | AUC   | T@1%F  |
> | No Attack               | 0.8032        | 0.2620 | 0.7532       | 0.1920 | 0.9800| 0.7780 | 0.7753| 0.1240 | 1.0000 | 1.0000 | 1.0000 | 0.9980 | 0.6980         | 0.3540 | 0.7252| 0.2400 |
> | Suggested Baseline       | 0.6972        | 0.0640 | 0.6199       | 0.0240 | 0.9531| 0.4280 | 0.8446| 0.0680 | 0.7097 | 0.0700 | 0.8807 | 0.3620 | 0.8030         | 0.1780 | 0.6836| 0.0340 |
> | AdvPara (RADAR)         | 0.5495        | 0.0260 | 0.4810       | 0.0000 | 0.8273| 0.1740 | 0.7340| 0.0220 | 0.7487 | 0.0780 | 0.7859 | 0.2380 | 0.4667         | 0.0140 | 0.4213| 0.0020 |
> | AdvPara (RoBERTa-Large) | 0.1447        | 0.0000 | 0.3354       | 0.0000 | 0.7794| 0.1460 | 0.7641| 0.0440 | 0.7682 | 0.0920 | 0.8339 | 0.2560 | 0.3510         | 0.0060 | 0.3932| 0.0000 |
> | AdvPara (RoBERTa-Base)  | 0.5590        | 0.0140 | 0.1174       | 0.0000 | 0.8679| 0.2840 | 0.8275| 0.0620 | 0.7627 | 0.1000 | 0.8475 | 0.2860 | 0.4963         | 0.0240 | 0.4697| 0.0040 |
> | AdvPara (MAGE)          | 0.5501        | 0.0420 | 0.4310       | 0.0020 | 0.5291| 0.0420 | 0.8089| 0.0660 | 0.7471 | 0.0960 | 0.8340 | 0.3020 | 0.2974         | 0.0120 | 0.3267| 0.0060 |
>
> Our findings show that our method—guided by a signal from the guidance detector—consistently outperforms the suggested baseline in terms of attack effectiveness in most cases. However, it's important to note that this baseline assumes knowledge of the deployed target detector, as it relies on using that same detector to select the most evasive paraphrase for itself. This represents a best-case scenario for attacking any particular detector, and relies on stronger assumption of having multiple query access to the target detector, which may not be practical. Even under this overly favorable condition for the suggested baseline, our method continues to outperform it.
>
>
> ### What happens if the guidance is not applied at all decoding steps?
> Thank you for another insightful question. To address this, we add a new set of ablation experiments by only applying detector guidance at every N steps during auto-regressive generation of the paraphraser. That is, the paraphraser in this case generates N-1 tokens as how a simple paraphraser would do, and then sample the N$^{th}$ token adversarialy guided by the detector. Here we use OpenAI-RoBERTa-Large as the guidance detector and run adversarial paraphrasing on 500 randomly selected texts. The resulting mean AUC and TPR@1%FPR across all 8 deploy detectors are shown below.
>
> | method                          | mean AUC | mean T@1%F |
> |---------------------------------|----------|------------|
> | No Attack                       | 0.8418   | 0.4935     |
> | Simple Paraphrase              | 0.8588   | 0.2885     |
> | AdvPara (RoBERTa-Large) (N=1)       | **0.5500**   | **0.0740**     |
> | AdvPara (RoBERTa-Large) (N=2)       | 0.6632   | 0.1333     |
> | AdvPara (RoBERTa-Large) (N=3)       | 0.7266   | 0.1690     |
> | AdvPara (RoBERTa-Large) (N=4)       | 0.7593   | 0.1820     |
> | AdvPara (RoBERTa-Large) (N=5)       | 0.7788   | 0.2115     |
>
> It can be seen that applying the detector guidance at every decoding step leads to the most effective attack, and the attack effectiveness degrades as N becomes larger. However, even with N=5, AdvPara still shows stronger attack effectiveness compared to simple paraphrasing baseline. Using a larger N can also reduce the attack latency by utilizing less detector guidance.
>
>
>
> ## Intuition on the transferability of our attack
> We appreciate your concern and would like to offer the following clarification.
> Our intuition, as explained in lines 195-201 of the paper, is that detectors converge to a shared underlying human text distribution with the aim of minimizing false positives and false negatives. Hence we can use one detector to guide the paraphrasing process for evading other detectors that utilizes the same underlying human distribution.
> However, since the true human text distribution is inherently intractable, we cannot directly verify how closely each detector’s learned distribution aligns with it. While MAGE may show the highest AUC on this particular dataset, this does not necessarily imply that it has best approximated the true human distribution. Based on our intuition, the intuitively most effective guidance detector for transferability would be the one whose learned human distribution shares the greatest average overlap with those of the deployed detectors. That said, this remains just an intuition, as neither the true human distribution nor the internal distributions learned by these detectors are directly observable or measurable.
>
>
>
> ## Would RADAR be able to detect the proposed attack if it was included in their training data?
> Thank you for raising this meaningful point. In short, yes that is possible, but such training may not be necessarily beneficial.
> Being able to contribute to generating adversarial datasets for improving the robustness of trained detectors via adversarial training is a part of our contributions. However, it's worth noting that in AI text  detection, there is a trade-off between improving type-I and type-II errors. As one optimizes to improve type-I errors, type-II errors will go worse. This was theoretically proven by Corollary 3 in [1]. Therefore, it is not guaranteed that training detectors with such an adversarial dataset will help make them truly robust.
>
>
> [1] Sadasivan et al. "Can AI-generated text be reliably detected?", TMLR 2025.

---

> ### Author Response · Authors · 2025-08-06
>
> Dear Reviewer nwS2,
>
> As we are approaching the end of the discussion period, we would like to kindly inquire whether you have further questions regarding our work.
>
> To summarize, during the rebuttal, we demonstrated the necessity of applying detector guidance by comparing it with the newly suggested additional baseline, and we also ablated our results by exploring the idea of applying guidance at every $N^{\text{th}}$ decoding step. Additionally, we have provided clarifications regarding our intuition and the potential for adversarial training.
>
> Your suggestions have greatly helped improve our paper, and we will diligently incorporate the new experiments and clarifications into the updated version. We hope that our new results and clarifications have helped address your questions. If so, we’d sincerely appreciate it if you would consider uprating our work. Your recognition means a great deal to us. If you have any remaining questions, we would be more than happy to address them.
>
> Thank you once again for your valuable time and insights!
>
> Best regards,
> The Authors

---

> > ### Comment · Reviewer_nwS2 · 2025-08-08
> > **Official Comment by Reviewer nwS2**
> >
> > Thank you for your detailed response. Your response addressed most of my concerns. I will maintain my positive score.

---

> > > ### Author Response · Authors · 2025-08-08
> > >
> > > Dear reviewer nwS2,
> > >
> > > Thank you very much for your reply. As we are only a few hours away from the end of the discussion period, we would like to kindly ask if you have any remaining fundamental concerns regarding our paper. We would be happy to address any questions or clarifications that might help increase your confidence in our work, and potentially support a higher evaluation.
> > >
> > > Thank you once again for your invaluable time and feedback!
> > >
> > > Best regards,
> > > Authors

---

### Official Review · Reviewer_f6cw · 2025-07-03

**Clarity:** 3
**Significance:** 2
**Originality:** 2
**Rating:** 4
**Confidence:** 3

**Summary:**

The paper introduces a framework designed to evade AI-generated text detectors by transforming machine-generated content into human-like text. The authors propose Adversarial Paraphrasing, a training-free attack that leverages an instruction-tuned LLM to paraphrase text while being guided by an AI text detector to optimize for evasion. Experiments demonstrate that adversarial paraphrasing significantly reduces detection rates compared to baseline methods like simple or recursive paraphrasing. The attack maintains high text quality, as evaluated through perplexity scores and automated GPT-4o ratings. It also provides empirical evidence of the attack’s universality.

**Questions:**

see above "Weaknesses" part.

**Ethical Concerns:**

["NO or VERY MINOR ethics concerns only"]

**Final Justification:**

The author provides additional empirical results, which address my concerns.

**Limitations:**

yes.

**Quality:**

3

**Strengths And Weaknesses:**

1.	Overall, the paper is well-written. My main concern lies in the claimed trade-off between text quality and detector evasion. First, the definition of “text quality” is somewhat vague. Is it about preserving semantic similarity to human-written text, linguistic fluency, or other dimensions? Fundamentally, text quality and detector evasion are not necessarily conflicting objectives. for example, a human writer can produce high-quality text that is also recognized as human-written by detectors. Therefore, I find the premise of this trade-off unclear in the current context and suggest that it be clarified more explicitly.

2.	In Figure 5, the lower part is difficult to interpret. The authors conclude that “simple paraphrases outperform adversarial paraphrases less than half of the time,” but this is not immediately evident from the figure. There appear to be three categories: noadv wins, ties, and adv wins. Does "noadv" refer to “simple paraphrases”? If so, this should be clearly stated to aid interpretation.

3.	The proposed method involves iterative, detector-guided token selection, which likely increases computational cost. The paper does not include a detailed efficiency analysis, for example, in terms of latency or resource requirements, especially for large-scale deployment scenarios.

4.	The effectiveness of adversarial paraphrasing depends heavily on feedback from a specific detector. It would be useful to discuss how the method would generalize if future detectors adopt fundamentally different detection mechanisms, which may render the current approach ineffective.

---

> ### Author Rebuttal · Authors · 2025-07-31
>
> We sincerely appreciate your thoughtful and detailed feedback. We are glad that you find our paper well-written. We address all your comments below.
>
> ## Trade-off between text quality and detector evasion
> ### Definition of text quality
> Thank you for asking this. The definition of “text quality” indeed encompasses multiple dimensions. In our evaluation, we focused on **both semantic equivalence to the original text** and **clarity, fluency, and naturalness of the text itself** (as captured in the GPT-4o quality ratings, win-rate, and perplexity analyses). We will clarify this in our revised draft.
>
> ### Is there necessarily always a trade-off between detection evasion and text quality?
> Thank you for raising this important question. As rightly pointed out by the reviewer, detection evasion and text quality may not be inherently conflicting objectives. Specifically, if quality is measured in terms of perplexity, then yes, there is a trade-off, as our method selects from the top-k tokens that minimize the guidance detector score rather than those with the highest log-probability scores. However, if quality is assessed based on human perception (i.e., alignment with the "gold standard" of fluency and coherence), then one may not necessarily need to trade quality for evasion. As demonstrated by our GPT-4o quality ratings, adversarial paraphrases were rated 4 or 5 out of 5 in 87% of cases—averaged across three datasets and four guidance detectors. This shows that it is certainly possible to evade detection without compromising text quality in many cases. However, we acknowledge that our method, though only in a small proportion of cases, results in reduced text quality. This is why we note this trade-off as a more general finding.
>
> ## Figure 5 labeling
> We apologize for the confusion. Yes, “noadv” refers to simple paraphrasing without detector guidance. We will correct this in our revised draft.
>
> ## Efficiency and latency
> Thank you for the suggestion. We performed adversarial paraphrasing across 100 random text samples, for 5 trials per setting, with all four guidance detectors. To simplify measurement and report time on a per-sample basis (rather than per batch), we used a batch size of 1 for these trials. In practice, including in the main experiments throughout the paper, we use larger batch sizes, which significantly reduce the overall time cost. The average paraphrasing time per sample across 5 trials is reported below as the mean ± standard deviation (in seconds):
>
> - Simple paraphrasing: 7.18 ± 0.13
> - Guidance=RADAR: 9.69 ± 0.20
> - Guidance=MAGE: 16.71 ± 0.74
> - Guidance=RoBERTa-Large: 10.20 ± 0.18
> - Guidance=RoBERTa-Base: 8.64 ± 0.11
>
> As shown, most guidance detectors introduce relatively low latency compared to simple paraphrasing. The higher latency observed with MAGE is due to its LongFormer-based architecture, which incurs longer inference time than the RoBERTa-based models.
>
> **Overall, the latency is primarily determined by the complexity of the guidance detector model itself.** From a computational cost perspective, the detector adds minimal overhead in terms of FLOPs compared to the paraphrasing LLM, which dominates the total computation. In our experiments, the paraphrasing LLM contains ~8 billion parameters, whereas the detectors only contain 100–350 million parameters—amounting to less than 5% of the paraphraser’s size for the larger detectors, and less than 2% for the smaller ones.
>
> It is also worth noting that the exact runtime per sample may vary based on the sequence length, which is inherently dependent on the length of the input text. For the sake of fair comparison—to ensure that AdvPara produces a similar number of tokens as simple paraphrasing—we report the mean and standard deviation of the token count for the texts:
>
> | Text                             | Mean Token Count | Std Dev |
> |----------------------------------|-----------------|---------|
> | Original texts                    | 173.73          | 38.46   |
> | Simple Paraphrase                 | 170.81          | 39.69   |
> | AdvPara (RoBERTa-Base)            | 175.30          | 51.27   |
> | AdvPara (RoBERTa-Large)           | 171.25          | 45.39   |
> | AdvPara (RADAR)                   | 169.68          | 60.07   |
> | AdvPara (MAGE)                    | 164.18          | 54.39   |
>
> ## Generalization to future detectors with different detection mechanisms
> Thanks for raising this concern. We would like to make the following clarification.
>
> Intuitively, our generalization capability comes from the intuition that, as explained in lines 195-201 of the paper, detectors converge to learn a shared underlying human text distribution with the aim of minimizing false positives and false negatives. And this shall hold regardless of the specific mechanism of the detector.
>
> Empirically, our experiments have demonstrated the generalization ability of our method across:
> - other neural network-based detectors,
> - watermark-based detectors, and
> - zero-shot detectors.
>
> The latter two represent fundamentally different detection mechanisms from the guidance detector used during paraphrasing. This supports our claim that the method generalizes beyond a single detection paradigm. Hence we use these transferability results to claim that our attack could work on a future detection mechanism that utilizes the same underlying human text distribution.

---

> > ### Comment · Reviewer_f6cw · 2025-08-04
> > **Reply**
> >
> > Thank you to the authors for the clarification.
> >
> > I appreciate the additional empirical results, which address most of my concerns.

---

> > > ### Author Response · Authors · 2025-08-06
> > >
> > > Thank you very much for your reply! We’re very glad to have addressed your concerns. Your suggestions have greatly helped improve our paper, and we will diligently incorporate the new experiments and clarifications into the updated version. Based on these updates, we would sincerely appreciate it if you could consider uprating our work. Your recognition means a great deal to us. If you have any remaining questions, we would be more than happy to address them. Thank you again for your valuable time and insightful comments.

---

> > > ### Author Response · Authors · 2025-08-08
> > >
> > > Dear reviewer f6cw,
> > >
> > > Thanks again for your helpful comments. As we are only a few hours away from the end of the discussion period, we would like to kindly ask if you have any remaining fundamental concerns regarding our paper. We would be happy to address any questions or clarifications that might help increase your confidence in our work, and potentially support a higher evaluation.
> > >
> > > Thank you once again for your invaluable time and feedback!
> > >
> > > Best regards,
> > > Authors

---

### Note · Authors · 2025-08-11

We sincerely appreciate all reviewers for their time and feedback on our submission. We are glad to see that the reviewers found our paper to be well written (f6cw, TeXh), our proposed attack to be creative (nJqj), effective (nwS2, TeXh, nJqj), and transferable (nwS2), and our evaluation to be comprehensive (nwS2, TeXh, nJqj). We are especially encouraged that all reviewers unanimously gave a positive recommendation in recognition of our work.

During the rebuttal phase, we carefully considered all raised comments and responded with a detailed and comprehensive rebuttal. We'd like to use this final remark opportunity to give a brief summary and overview of the common questions and our responses. Full details are in the respective threads.

**Common questions:**

- **Latency and overhead?** (f6cw, nJqj) We conducted further experiments to show that the latency is small and acceptable.
- **Text quality?** (f6cw, TeXh, nJqj) We clarified several constraints applied during decoding to ensure high-quality output. We also pointed to our extensive quality analysis in Section 5, along with many text examples in Table 1 in the paper and all tables in Appendix D, and added a fourth quality metric as suggested by TeXh. All results show that in most cases the texts remain coherent and fluent.
- **Stylistic “human-likeness”?** (TeXh, nJqj) We clarified that the goal of this work is not to perform style transfer, but to propose an effective and transferable attack for defeating AI-text detection. We also showed human texts are extremely diverse, such that one cannot define a single stylistic representation that reliably characterizes all human-generated texts.
- **Hyperparameter choices?** (TeXh, nJqj) We conducted additional ablations to show the effect of $p$ and $k$ on our attack, and to explain the hyperparameter choices used in the paper.

Together with our individual responses to reviewer-specific points, we are confident we have resolved most—if not all—concerns.

Thank you again to all reviewers for their valuable feedback, which has greatly helped improve our work. We will diligently incorporate the new experiments and clarifications into the updated manuscript.

---

### Decision · Program_Chairs · 2025-09-17

**Decision:**

Accept (poster)

**Comment:**

This paper introduces Adversarial Paraphrasing, a training-free framework that humanizes AI-generated text by guiding an LLM with an AI text detector. Reviewers liked the creativity, strong empirical effectiveness, and broad transferability of the attack across diverse detection systems. The comprehensive experimental evaluation was also a noted strength.

As far as weaknesses, the reviewers pointed some out, but in basically all cases, the authors responded with rebutall that satisfied the concerns. I would encourage the authors to use these additional experiments and analaysis to strength the final paper.

Some of these included:
1) Questioning the definition of "text quality" and the claim of "humanizing" text. Authors clarified their quality metrics (GPT-4o, win-rate, perplexity), added semantic similarity, provided more examples, and revised "human-likeness" terminology which addressed these concerns.
2) Efficiency: Authors provided empirical latency measurements, explaining the low overhead relative to the paraphraser LLM.
3) Baseline Comparisons: A reviewer suggested a "pick-best-of-N paraphrases" baseline. Authors added this experiment, demonstrating their method's better performance even against this strong baseline.
Ablation Studies & Failure Cases: Reviewers requested more hyper-parameter ablations and analysis of failure cases. Authors provided these
4) Necessity of per-step guidance: A reviewer questioned why guidance was needed at each decoding step. Authors provided an ablation showing that per-step guidance is indeed more effective than less frequent guidance.

The authors were very on-the-ball in addressing the concerns raised by reviewers, who consistently thought highly of the work. The authors provided strong empirical evidence for the attack's effectiveness, and the work is useful for highlighting vulnerabilities in current detection strategies and will motivate more research in this area as well as in defenses against these attacks.